# 1 Monitoring CO2 in diverse European cities: Highlighting needs and

# 2 challenges through characterisation

- Ida Storm<sup>1,2</sup>, Ute Karstens<sup>1</sup>, Claudio D'Onofrio<sup>1</sup>, Alex Vermeulen<sup>1</sup>, Samuel Hammer<sup>3</sup>, Ingrid Super<sup>4</sup>,
- Theo Glauch<sup>3,5</sup>, and Wouter Peters<sup>2,6</sup>
- <sup>1</sup>ICOS ERIC Carbon Portal, Department of Physical Geography and Ecosystem Sciences, Lund University, Lund, 22362,
- Sweden
- <sup>2</sup>Environmental Sciences Group, Wageningen University, 6700 AA Wageningen, the Netherlands
- <sup>3</sup>Institute of Environmental Physics, Heidelberg University, Heidelberg, 69117, Germany
- <sup>4</sup>Department of Air Quality & Emissions Research, TNO, 3508 TA Utrecht, the Netherlands
- 5 Institut für Physik der Atmosphäre, Deutsches Zentrum für Luft- und Raumfahrt, Oberpfaffenhofen, 82234, Germany
- <sup>6</sup>Centre for Isotope Research, University of Groningen, 9747 AG Groningen, the Netherlands
- Correspondence to: Ida Storm (ida.storm@icos-cp.eu)
- Abstract. For the development of a joint European capacity for monitoring CO<sub>2</sub> emissions, we created the framework "CO<sub>2</sub>
- Monitoring Challenges City Mapbooks v1.0" (CMC-CITYMAP). It includes a Jupyter notebook tool (Storm et al., 2025a,
- https://doi.org/10.18160/P8SV-B99F) which we use to characterise and cluster cities based on aspects relevant for different
- CO<sub>2</sub> monitoring challenges. These include:
- (a) determining background levels of CO<sub>2</sub> inflow into a city ("background challenge").
- (b) separating the anthropogenic emissions from the influence of the biosphere ("biogenic challenge").
- (c) representing spatially and temporally non-uniform emissions in models ("modelling challenge").
- (d) implementing observation strategies not covered by the other challenges ("application-specific observational challenge").
- We provide and discuss the challenges on a city-by-city basis. Our primary focus, however, is on the relationships between
- 25 cities: best practices and lessons learned from monitoring CO<sub>2</sub> emissions in one city can be transferred to other cities with
- similar characteristics. Additionally, we identify cities with characteristics that strongly contrast with those of cities with
- existing urban monitoring systems.
- While the notebook tool includes 308 cities, this paper focuses on the results for 96 cities with more than 200,000 inhabitants.
- We place a particular emphasis on Paris, Munich, and Zurich. These cities are pilot cities for the Horizon 2020-funded project
- Pilot Application in Urban Landscapes ("ICOS Cities"), where a range of urban CO<sub>2</sub> monitoring methods are being
- implemented and assessed. According to our analyses, Zurich—and Munich especially—should be less challenging to monitor

than Paris. Examining the challenges individually reveals that the most significant challenge relative to the other cities is the "modelling challenge" (c) for Zurich and Paris. Complex urban topography adds to the challenge for both cities, and in Zurich, the natural topography further amplifies the challenge. Munich has low scores across all challenges, but with the greatest challenge anticipated from the "application-specific observational challenge" (d). Overall, Bratislava (Slovakia) and Copenhagen (Denmark) are among the most distant from Paris, Munich, and Zurich in our dendrogram resulting from numerical cluster-analysis. This makes them strong candidates for inclusion in the ICOS Cities network, as they would potentially provide the most information on how to monitor emissions in cities that face different challenges.

### 1 Introduction

"Cities are where the climate battle will largely be won or lost", stated United Nations Secretary-General António Guterres at the 2019 C40 Mayors Summit. In 2020, cities accounted for approximately 67-72% of global CO<sub>2</sub>-equivalent emissions based on consumption-based accounting (Lwasa et al., 2022). This share will only increase as the urban population is projected to rise from 4.2 billion in 2018 to 6.7 billion by 2050 (United Nations, 2018). In response, many cities in Europe are committed to the EU's climate targets to achieve net-zero emissions by mid-century (European Commission, 2019). Often, they have joined forces in their efforts through initiatives such as C40 Cities (C40 Cities, n.d.) and the Covenant of Mayors (Covenant of Mayors, n.d.), as well as inclusion in the European Union's mission "100 Climate-Neutral and Smart Cities by 2030" (European Commission, Directorate-General for Research and Innovation, 2024). Cities can achieve climate neutrality by ensuring they remove as much greenhouse gases as they emit. To reach this goal, they have drawn up and committed themselves to implement climate action plans with various mitigation efforts. However, many cities lack the detailed and timely information on their emission history and trends, which is necessary to evaluate effective action (Hsu, 2020). While various options exist for obtaining this information, verifying emissions always requires actual observations. Determining the most effective strategies for these observations is an active area of research.

Most cities that engage in emission monitoring use "bottom-up" approaches that usually do not include direct observations: activity data (such as traffic counts) are combined with emission factors (such as kgCO<sub>2</sub>/vehicle km (vkm)), and the sophistication of its implementation varies. Several public protocols are available for cities to develop self-reported inventories (SRIs), including those from ICLEI - Local Governments for Sustainability, and the Global Covenant of Mayors (ICLEI, n.d.; Global Covenant of Mayors, 2023). The estimates resulting from using different protocols can show significant differences (e.g., Albarus et al., 2023; Gurney et al., 2021; Gately and Hutyra 2017; Lian et al., 2023). For example, Gurney et al. (2021) found an average under-reporting of 18.3% when comparing the annual emission estimates for 48 U.S. cities with local SRIs to the common inventory "Vulcan". The latter has shown consistency with observations in previous studies (Gurney et al., 2020; Lauvaux et al., 2020; Basu et al., 2020). The range of under- or over-reporting spanned from -145.5% to +63.5%. Uncertainties become even larger when estimating emissions at higher spatial and temporal resolutions (Super et al., 2021).

For example, Lian et al. (2023, Fig. S10) showed particularly large discrepancies in individual 1 km<sup>2</sup> grid cells when comparing two emission inventories.

In the "top-down" approach, various types of observations are used to verify and potentially refine the emission estimates. The observational methods available include—but are not limited to—measuring concentrations using sensors of varying accuracy and precision, observing total column concentrations through surface-based remote sensing and satellites, and measuring direct fluxes with eddy covariance. However, these observations concern total CO<sub>2</sub> and to isolate the fossil fuel component, different types of observations should be used. Options include measurements of co-emitted trace gases such as CO (e.g. Turnbull et al., 2006; Nathan et al., 2018) and NO<sub>x</sub> (e.g. Lopez et al., 2013); co-located trace gases such as SF<sub>6</sub> (e.g. Turnbull et al., 2006; Turnbull et al., 2011); and isotopes like <sup>14</sup>C in CO<sub>2</sub> (e.g. Turnbull et al., 2006; Lopez et al., 2013; Miller et al, 2020). There are several options for using observations to provide information on emissions, often synergistically to improve each other (Miles et al., 2021). A comprehensive account of the options can be found in IG31S "Urban Emission Observation and Monitoring Good Research Practice Guidelines" (World Meteorological Organization, 2025). To produce spatially explicit maps with adjusted (bottom-up) emissions, inverse modelling is commonly applied. This approach relates the observations, or the observed upwind-downwind gradients (e.g. Bréon et al., 2015, Super et al., 2017; Staufer et al., 2016), to CO<sub>2</sub> exchanges within the city using transport models. Next, the CO<sub>2</sub> emissions are optimised to fit better with the observations. There are uncertainties also in the adjusted emissions, and a study period of at least a few years may be required to confirm a trend in the emissions with high confidence (Lauvaux et al., 2020).

Several factors make monitoring CO<sub>2</sub> emissions particularly challenging and prone to uncertainties. Based on our experiences and a literature review of monitoring efforts in cities, we have identified four main areas of challenges. The first is to accurately represent the variability in boundary conditions, meaning the "background" concentration of air flowing into the city (the "background challenge"). This can significantly affect the results as the increase in concentrations from city emissions is relatively small, even for large cities. For example, in Indianapolis the enhancement at the downwind site was only about 3 ppm in October 2012 (averaged over 17-22UTC), according to Lauvaux et al. (2016). Using only models to represent the background can introduce errors that are larger than this enhancement, with Lian et al. (2021) reporting differences as large as 5 ppm for background concentration for Paris between two models. In addition, this may create seasonal biases (Sargent et al., 2018). The alternative is to use observations, which comes with the challenge of selecting spatially representative locations that have limited local flux contributions and well-understood atmospheric dynamics (Sargent et al., 2018). Seemingly homogeneous land cover classified as "cropland" may require extra attention, as the associated fluxes can vary significantly due to different management practices and crop rotation cycles. For example, in Miles et al., (2021) two background towers classified as "agricultural" gave significantly different values.

A second challenge is correctly attributing the fossil fuel CO<sub>2</sub> (ffCO<sub>2</sub>) component in observed total CO<sub>2</sub> (the "biogenic challenge"). Correlated and co-emitted trace gases, as well as <sup>14</sup>C in CO<sub>2</sub>, have already been mentioned as useful observations for this purpose. They can be used to optimise modelled prior biogenic fluxes in addition to the anthropogenic emissions (e.g. Miller et al., 2020). Historically, biogenic flux models have been unable to resolve urban vegetation and its associated fluxes. For example, Lian et al. (2023) found that their biogenic model only resolved the two largest parks within the Île-de-France region. They were not optimising the biogenic fluxes and instead saw large adjustments to their prior ffCO<sub>2</sub> emissions, especially during the growing seasons. One alternative strategy has been to study only the dormant season and assume biogenic exchange to be insignificant (e.g. Lauvaux et al., 2016). Recent developments, including Urban-VPRM (Hardiman et al., 2017) and pyVPRM (Glauch et al., 2025), can better resolve sub-kilometre patches of vegetation. Their improvements stem mainly from the use of high-resolution satellite products, but they are still parametrised with rural flux data and assumed to function in the same way also in urban areas. However, urban management practices have been shown to violate this assumption. For example. Smith et al. (2019) found that urban trees have growth rates up to four times compared to those observed in a nearby forest. A more recent study from Havu et al. (2024), describes a significant CO<sub>2</sub> uptake for the city of Helsinki. This may be attributed to higher ambient CO2 mole fractions, nutrient variability, and water availability from irrigation. However, there are also urban studies where lower CO<sub>2</sub> uptake and decreased productivity are observed due to factors such as pollutant loads or poor soil conditions (Roman and Scatena 2011; Ainsworth et al. 2012). Correctly representing these responses in biogenic flux models is especially important when the biogenic component is large compared to anthropogenic emissions. Many studies have reported estimates for the biogenic component, with its relative contribution varying to large extent depending on the city, season, and time of the day (e.g., Turnbull et al., 2015; Gurney et al., 2017; Sargent et al., 2018; Winbourne et al., 2022). Studies in Boston (Sargent et al., 2018) and the Washington, DC/Baltimore area (Winbourne et al., 2022) found that the influence of biogenic fluxes on the city's net flux was sometimes comparable to that of anthropogenic emissions.

The third challenge is representing the urban carbon landscape in models (the "modelling challenge"). While the biogenic fluxes are discussed separately, additional challenges arise from the highly non-Gaussian variability of emissions across both time and space. About 50% of fossil fuel related CO<sub>2</sub> emissions in Europe stem from large point sources, which are required to report their emissions under the EU ETS (European Union Emissions Trading Scheme) and the E-PRTR (European Pollutant Release and Transfer Register). Although many of these facilities report hourly emissions with high accuracy, most models are unable to use facility-specific data. Instead, they rely on standard temporal profiles to scale annual totals. This can introduce large uncertainties, as demonstrated in studies by Super et al. (2020; 2021). These uncertainties further increase near point sources, where representing the emission plume accurately is challenging due to the well-mixed assumption in most models (Lauvaux et al., 2016). Furthermore, as most emissions from point sources are released from a stack, models need to incorporate realistic vertical profiles (Brunner et al., 2019; Maier et al., 2022). Another challenge for transport models in the urban environment is to accurately represent airflow, which is complicated by variable topography and tall urban structures. There are models that can do this with some accuracy (e.g. Berchet et al., 2017; Gaudet et al., 2017), but they are

computationally expensive to run. For example, Berchet et al. (2017) use a catalogue-based approach where a set of precomputed steady-state flow and dispersion patterns is matched hourly to actual meteorological observations. These models require highly resolved spatiotemporal input data, including both biogenic fluxes and anthropogenic emissions.

The fourth challenge within the scope of this paper is the "application-specific observational challenge". Many challenges associated with implementing a basic observational network are inherently intertwined with the other discussed challenges. These include high precision CO<sub>2</sub> in-situ observations on tall towers, low- and mid-cost sensors, ground based total column FTIRs (Fourier Transform Infrared Spectroscopy), and eddy flux towers. In this fourth challenge, we include the challenges associated with the use of the isotope <sup>14</sup>C in CO<sub>2</sub> and making satellite observations. <sup>14</sup>C (radiocarbon), can be used to estimate the amount of ffCO<sub>2</sub> in a sample. However, high costs limit the spatial and temporal coverage of radiocarbon observations, and therefore co-emitted species such as CO are often used to fill the gaps. Calibration with co-located radiocarbon observations remains necessary. A key challenge with radiocarbon observations is accounting for the contribution to the atmospheric signal by radiocarbon emissions from nuclear facilities (e.g. Levin et al., 2003; Graven and Gruber, 2011; Bozhinova et al., 2014; Maier et al., 2023). The impact of these emissions depends on the proximity of sampling locations to nuclear facilities. Unaccounted emissions were estimated to mask about 15% of ffCO<sub>2</sub> emissions in flask samples collected at seven Integrated Carbon Observation System (ICOS) stations in the study by Maier et al. (2023). Even when considered in the ffCO<sub>2</sub> estimates, obtaining the appropriate temporal resolution for these emissions is difficult. This increases uncertainties in <sup>14</sup>C-based ffCO<sub>2</sub> estimates (Maier et al., 2023).

Another type of observation considered within the scope of the fourth challenge is column-averaged CO<sub>2</sub> dry air mole fraction (XCO<sub>2</sub>) from satellites. These observations require a clear sky for accurate overpass measurements which can significantly limit the number of samples collected. For example, in a synthetic study for Berlin, Kuhlmann et al. (2019) found that out of the 365 days in 2012, only 50 appeared suitable to observe the CO<sub>2</sub> plume from space due to unfavourable meteorological conditions during the other 315 days. Furthermore, the emissions during the sample collection were 18% higher than the annual total for Berlin, requiring temporal profiles to correct for this sampling bias. However, as shown in Super et al. (2020), temporal profiles come with sometimes large additional uncertainties. Yet another challenge with satellite observations is that only large emissions provide a sufficient signal-to-noise ratio in observed XCO<sub>2</sub> enhancement. Wang et al. (2020) suggested that emissions from a city or a power plant larger than 7.33 MtCO<sub>2</sub> yr<sup>-1</sup> (2 MtC yr<sup>-1</sup>) could potentially be constrained between 8:30 and 11:30 using the CO2M instrument, which has a planned launch in 2026. The threshold corresponds to a posterior uncertainty smaller than 20% for more than 10 times within a year.

In this study, we quantify and compare the challenges for 96 cities by relating them to information gathered from relevant spatial data layers. This is done using various Geographical Information Science (GIS) techniques to condense information from multiple data layers into 18 city metrics. These metrics represent specific characteristics of the city and are weighted

based on factors that are deemed to make emission monitoring challenging. Each city is presented in individual "mapbooks", which show their results and associated maps. These mapbooks can be used by stakeholders or local experts, as well as in national or pan-European monitoring strategies, including ICOS (Integrated Carbon Observation System) and Copernicus' monitoring and verification system (MVS). The full framework of CMC-CITYMAP also includes an interactive Jupyter notebook that can be downloaded or run on the ICOS Jupyter service. It allows users to update the analyses presented in this study and explore additional available metrics.

After an overview of our study area and selected cities (Sect. 2.1), Sect. 2.2 explains how the spatial information layers are reduced to metrics. Sect. 2.3 connects the metrics to the four monitoring challenges and Sect. 2.4 is detailing how they are integrated into challenge scores and used in further analyses. Next, the results are presented in four sections. They begin with the characteristics of individual cities (Sect. 3.1), proceed to city comparisons (Sects. 3.2 and 3.3), and conclude with a cluster analysis focusing on the implications for a joint European urban monitoring capacity. A discussion of the results follows (Sect. 4), and the study is concluded in Sect. 6. Section 5 provides links to relevant resources for the study, including its associated Jupyter notebook tool and mapbooks.

2 Methods

Spatial information layers representing city characteristics relevant to the monitoring challenges have been selected. These layers come from various sources and are available at different resolutions (see Table 1). City borders are used to subset the layers, and statistical properties or derived indices are then used to generate comparable metrics for each city. In some cases, multiple layers are combined to create a single metric, such as the ratio between biogenic uptake and anthropogenic emissions (see Sect. 2.2.3). When applicable, the selected time period is the dormant season during daytime, which helps reduce the influence of the biosphere and usually means well-mixed conditions.

2.1 Cities and their surroundings

The city boundaries used in this study were downloaded from Eurostat's GISCO service (Eurostat, 2024). These include only cities within the European Union, and the delineation method follows the definition provided by the OECD (Organisation for Economic Co-operation and Development). Fine-grained population data was used to delineate urban centres, defined as contiguous areas of high population density (>1500 residents per km²) with a total population of at least 50,000 residents. In turn, the urban centres were associated with local administrative units, and if more than 50% of the population within a unit lived in the urban centre, the local administrative unit was defined as a city. In cases where adjacent local administrative units met the city criteria, they were merged to form a single city (Dijkstra et al., 2019).

A total of 308 cities in the European Union fall within our study region spanning from 2°W to 19°E and from 47°N to 56°N. This is the area where one of our key data sources—the high-resolution emission data from TNO (the Dutch Organization for Applied Scientific Research)—is available. For our study we have considered only cities with over 200,000 inhabitants, resulting in the 96 cities depicted in Fig. 1. Most are found in Germany (43), the Netherlands (15), France (13) and Poland (9). The surroundings of the cities are defined as the buffer area extending 20 km beyond the city boundaries and is used for some of the metrics. Additionally, the surrounding buffer area in the dominant 30-degree wind direction is used to subset data for separately weighted metrics (see Table 1; Sect. 2.2.1). This puts higher significance on the characteristics of the area upwind of the city.

Figure 1: Overview of the 96 characterised cities. The points represent cities, and their colours indicate which of the four challenges has the highest score. The size of the points increases with the anticipated overall challenge to monitor emissions in them after weighing the individual challenges equally (see Sect. 2.4.1).

#### 2.2 Extraction of city metrics

Table 1 lists all the input datasets along with brief information on how they are analysed to derive metrics for the cities, which are used in further analyses. The datasets are available for the entire region, which is a prerequisite for making comparisons across the 96 cities. Alternative datasets and derived metrics—which were excluded from this study—are also available in the

notebook tool (Storm et al., 2025a). Sections 2.2.1 through 2.2.6 focus on the datasets and how they are used to derive the metrics. Sections 2.3.1 through 2.3.4 motivate how the data layers are associated with the individual challenges to estimate their relative difficulty. Finally, sections 2.4.1 through 2.4.4 outlines how the metrics are integrated and analysed. This includes how the weights (column "Challenge (weight)" in Table 1) are applied to the individual metrics, and how the metrics are adjusted so that higher values consistently correspond to a greater monitoring challenge before they are combined.

| Data (section)                                       | Resolution        | Reference<br>year data | Reference                                                    | Metric                                                                                                                        | Metric implication                                        | Challenge<br>(weight)                      |
|------------------------------------------------------|-------------------|------------------------|--------------------------------------------------------------|-------------------------------------------------------------------------------------------------------------------------------|-----------------------------------------------------------|--------------------------------------------|
| Wind (2.2.1)                                         | 0.25° x 0.25°     | 2018                   | ERA5 reanalysis<br>(Hersbach et al.,<br>2023)                | Fraction of time wind from the dominant wind direction (limited to wind speed >2 m s <sup>-1</sup> )                          | Expected constancy of concentration footprint             | Background (30%)*                          |
|                                                      |                   |                        |                                                              | Fraction of time with wind speed >2 m s <sup>-1</sup>                                                                         | Stagnant flow conditions                                  | Background (10%)*                          |
| emissions by<br>sector and<br>source type<br>(2.2.2) | 1/60° x<br>1/120° | 2018                   | TNO high resolution emission inventory (Kuenen et al., 2022) | Emission intensity<br>buffer Emission intensity<br>buffer dominant<br>wind direction<br>(limited to >2 m<br>s <sup>-1</sup> ) | Non-city<br>emissions within<br>the expected<br>footprint | Background (20%) Background (20%)          |
|                                                      |                   |                        |                                                              | Share point source<br>emission  Non-point-source<br>emission spatial                                                          | Expected ffCO <sub>2</sub> signal                         | Modelling (30%) Modelling (20%)            |
| Land cover (2.2.3)                                   | 10m x 10m         | 2021                   | ESA<br>Worldcover v.2<br>(Zanaga et al.,                     | aggregation Vegetation heterogeneity                                                                                          | aggregation  Expected separation of biogenic signal       | Biogenic (30%)                             |
|                                                      |                   |                        | 2022)                                                        | Share cropland in buffer  Share cropland in buffer dominant wind direction (limited to >2 m s <sup>-1</sup> )                 | Non-city<br>cropland within<br>the expected<br>footprint  | Background<br>(10%)<br>Background<br>(10%) |
| Net Ecosystem<br>Exchange<br>(NEE) (2.2.3)           | 500m x<br>500m    | 2018                   | VPRM<br>(Mahadevan et<br>al., 2008;                          | NEE relative to ffCO <sub>2</sub>                                                                                             | Signal-to-noise potential of ffCO <sub>2</sub>            | Biogenic (40%)                             |
|                                                      |                   |                        | Glauch et al., 2025)                                         | Average NEE                                                                                                                   |                                                           | Biogenic (30%)                             |
| Building<br>height (2.2.4)                           | 100m x<br>100m    | 2018                   | GHSL: Global building heights                                | Average building height                                                                                                       | Expected complexity of                                    | Modelling (20%)                            |

| Landform (2.2.4)                                   | 90m x 90 m      | 2015 | (Pesaresi and<br>Politis, 2023)<br>Global SRTM<br>Landforms<br>(Theobald et al.,<br>2015)             | Share of flat areas                                                                    | urban topography Expected complexity of natural topography           | Modelling (15%)*                        |
|----------------------------------------------------|-----------------|------|-------------------------------------------------------------------------------------------------------|----------------------------------------------------------------------------------------|----------------------------------------------------------------------|-----------------------------------------|
| Topography (2.2.4)                                 | 25m x 25m       | 2011 | EU-DEM v1.1<br>(European<br>Environment<br>Agency, 2016)                                              | Topographic heterogeneity                                                              |                                                                      | Modelling (15%)                         |
| Emissions<br>from nuclear<br>facilities<br>(2.2.5) | Exact locations | 2021 | Annual emission totals of <sup>14</sup> CO <sub>2</sub> from nuclear facilities (Storm et al., 2024b) | Potential nuclear masking (see Eq. 1)  Nuclear sample selection bias                   | Expected interference of nuclear emissions when sampling radiocarbon | Observational (25%) Observational (25%) |
| Cloud cover (2.2.6)                                | 0.25° x 025°    | 2018 | ERA5 reanalysis<br>(Hersbach et al.,<br>2023)                                                         | Share of days with >30% cloud cover summer  Share of days with >30% cloud cover winter | Expected potential for satellite observations                        | Observational (25%) Observational (25%) |

<sup>\*</sup>For these contributions, a lower value means a greater monitoring challenge (see Sect 2.4).

Table 1: An overview of the different input data layers, the metrics they are used to derive, and the specific challenges they contribute to estimating. Their weights in their contributions to the challenges are provided as percentages. The weights within each category sum to 100%. For the overall challenge, the four individual challenges are equally weighted (see Sect. 2.4.1).

### 2.2.1 Wind

For the metrics related to wind, eastward and northward windspeed components at ten meters from European Centre for Medium-Range Weather Forecasts (ECMWF) Reanalysis v5 (ERA5; Hersbach et al., 2023) have been used. Data during daytime hours (09:00 to 18:00 UTC) in the winter months (January and February) of 2018 was used for the analyses. One of the derived metrics is the "Fraction of time with wind speed above 2 m s<sup>-1</sup>" at the centroid of the city boundary. The 2 m s<sup>-1</sup> threshold is also used to filter out low wind speeds when calculating the fraction of time the wind is from the dominant wind direction as well as in a couple of other metrics (see Table 1).

The dominant wind direction is determined by aggregating the wind direction into 30-degree bins, where north is defined as ranging from 345 to 15 degrees. The bin found to represent the dominant wind direction for a city is used in several metrics (see Table 1) to place emphasis on the surrounding area upwind of the city.

### 2.2.2 Anthropogenic CO<sub>2</sub> emissions

The bottom-up  $CO_2$  emission inventory used in this paper originates from TNO. It includes emissions from different sectors distributed on a  $1/60^{\circ} \times 1/120^{\circ}$  grid (approximately 1 km<sup>2</sup>). Emissions from power plants and industrial facilities are instead

assigned to their exact locations, as derived from input datasets including E-PRTR (Kuenen et al., 2022, Table 1). Standard temporal profiles (updated from Denier van der Gon et al., 2011) are applied to distribute the annual emissions into hourly values. This is done using sector-specific scaling factors for individual months, days of the week, and hours of the day. These profiles are used to get data comparable to biogenic activity at specific times (see Sect. 2.2.3).

All metrics related strictly to anthropogenic CO<sub>2</sub> emissions use the annual total for the year 2018. "Emission intensity buffer" uses emissions per km<sup>2</sup> in the 20-kilometer buffer area around the cities, with an additional metric, "Emission intensity buffer dominant wind direction", which highlights the upwind buffer area. "Share point source emissions" is the percentage of a city's total emissions that can be attributed to point sources.

In addition to metrics related to emission intensity and shares, there is a metric called "non-point-source emission spatial aggregation". It is defined as the share of the city's total area with the highest emission intensity that in combination holds 50% of the total emissions from non-point sources. Higher values therefore mean that remaining emissions are more evenly distributed in the city.

2.2.3 Biospheric CO<sub>2</sub> exchange

As a representation of biospheric CO<sub>2</sub> exchange, calculations of NEE (Net Ecosystem Exchange) provided by Heidelberg University were used. The calculations are based on a new implementation of the Vegetation Photosynthesis and Respiration Model (VPRM; Mahadevan et al., 2008) in the pyVPRM tool (Glauch et al., 2025). VPRM is a simple diagnostic model that uses remote sensing and meteorological data to estimate the NEE at high spatiotemporal resolution. This implementation uses MODIS Terra MOD09A1 Collection 6.1 8-day data (Vermote, E., 2021) at 500 m resolution and hourly ERA5 meteorological data with a resolution of 0.25 degrees to retrieve the two-meter temperature and the solar irradiance (Hersbach et al., 2023). In addition, land cover information from the Copernicus Land Cover Service is used at 100m resolution (Buchhorn et al., 2020).

For the metric related to the general biogenic activity in the city, average NEE at 15:00 UTC during winter (January and February) has been calculated. In the metric comparing NEE and emissions, the ratio between average city-wide NEE and anthropogenic ffCO<sub>2</sub> at 15:00 UTC in winter is used. If it is a challenge during this time of year, when the biosphere is dormant, it will also be a challenge during the rest of the year. To estimate how coherent the biogenic active areas are within the city, an "edge-to-area ratio" for vegetation is applied. Based on the European Space Agency (ESA) Worldcover dataset v2 (Zanaga et al., 2022), each 10m resolution cell attributed to vegetation (classes 10, 20, 30, 40, 90, and 100) is selected. "Edge cells" are defined as cells with at least one non-vegetated neighbouring cell. The final metric represents the fraction of vegetation cells that are classified as edge cells.

- The ESA Worldcover dataset is also used to include cropland information (class 40) in the metrics "share cropland buffer" and
- "share cropland in dominant wind direction". In both cases, 20-kilometer buffers around the cities are applied (see Sect. 2.1).
- For "share cropland buffer", the full buffer area is used. For "share cropland in dominant wind direction", only the buffer in
- the dominant wind direction is considered to emphasise the upwind area.

### 2.2.4 Natural and built-up topography

- The landform dataset by Theobald et al. (2015) is used to calculate the share of flat areas (classes 24 and 34) within the city,
- while average building heights are derived from the dataset by Pesaresi and Politis (2023). The building heights dataset has a
- 100m x 100m resolution, which is deemed adequate for a city-wide average and indicates whether the city has many tall
- buildings.

283

273

- The spatial variability in the natural topography—the "Topographic heterogeneity" metric—is captured by averaging the
- Terrain Ruggedness Index (TRI) for each 25m x 25m grid cell in the EU-DEM v1.1 (European Environment Agency, 2016).
- The TRI is calculated using the methodology outlined in Riley et al. (1999): each cell's value is determined by taking the
- square root of the squared and averaged elevation differences with its eight adjacent cells.

# 2.2.5 Radiocarbon (<sup>14</sup>CO<sub>2</sub>)

- When using <sup>14</sup>CO<sub>2</sub> observations to separate fossil and non-fossil contributions of urban CO<sub>2</sub> enhancements, it is essential to
- account for the impact of anthropogenic <sup>14</sup>C emissions from nuclear facilities. Nuclear emissions enhance the <sup>14</sup>C/C ratio
- masking part of the <sup>14</sup>C/C depletion due to the emission of ffCO<sub>2</sub>. This masking effect was on average 15% in flask samples
- collected at seven ICOS stations in the study by Maier et al. (2023). A Jupyter notebook hosted at the ICOS Carbon Portal
- (Storm et al., 2024a) is used to quantify nuclear masking using a modification of their Equation 2.3: ffCO<sub>2</sub> (C<sub>ff</sub>) is calculated
- using measured CO<sub>2</sub> ( $C_{meas}$ ) and  $\Delta^{14}C$  ( $\Delta^{14}_{meas}$ ), with and without considering the nuclear contribution ( $\Delta^{14}_{nuc}$ ). As in Levin et
- al. (2003), the relatively insignificant respiration term is excluded:

$$\Delta 14_{meas} = \frac{c_{bg} \cdot \Delta 14_{bg} + c_{meas} \cdot \Delta 14_{nuc} - 1000 \cdot c_{ff}}{c_{ff} + c_{bg}} \tag{1}$$

- $\Delta^{14}_{meas}$  is solved for based on modelled concentration timeseries calculated in the Jupyter notebook (Storm et al., 2024a;
- Karstens, 2023). The background concentrations ( $\Delta^{14}_{bg}$ ) are provided by the ICOS Radiocarbon Laboratory based on
- measurements from the Mace Head site in Ireland. Next, Eq. 1 is used once more to back-calculate what ffCO<sub>2</sub> (C<sub>ff</sub>) would
- need to be if the nuclear contribution term ( $\Delta^{14}_{\text{nuc}}$ ) was not considered. The result is compared to the original modelled ffCO<sub>2</sub>
- component (C<sub>ff</sub>) to calculate the impact of nuclear masking. For the calculation of the final metric for each city, the average
- differences in percent for January and February at 12:00 and 15:00 UTC are calculated.

Even when nuclear contributions are accounted for, they introduce additional uncertainties to the <sup>14</sup>C-based ffCO<sub>2</sub> estimates. This is primarily because the emission time profile—assumed to be flat and derived from annual nuclear emissions totals—does not accurately reflect the timing of emissions (Maier et al., 2023). This limitation motivates the current sampling strategy at the ICOS Radiocarbon Laboratory in Heidelberg: to avoid sampling when nuclear contribution exceeds 0.5 permil. A second metric, "nuclear sample selection bias", calculates the degree of sampling bias that could occur in cities if this observational monitoring strategy is adopted. The modelled concentration timeseries (12:00 and 15:00 UTC, January and February of 2021). It is subset to when the nuclear contribution is below 0.5 permil, based on calculations in the Carbon Portal notebook (Storm et al., 2024a). The metric is calculated as the percentage difference between the average ffCO<sub>2</sub> components in the subset and the average for the entire time series.

### 2.2.6 Cloud cover

Total cloud cover is extracted from ECMWF ERA5 at 12:00 UTC during the winter (January and February) and summer (June and July) of 2018. Winter and summer are included as separate metrics because cloud cover can exhibit significant seasonal variability depending on the city's location. 12:00 UTC was selected to match with the overpass time of the planned CO2M satellite mission (Kuhlmann et al., 2019). The 0.25 x 0.25 degree data cell in which each city falls is used to extract a time series of cloud cover in the individual cities. In turn, a threshold of 30% cloud cover is used to calculate the proportion of days when samples will likely need to be discarded.

# 2.3 Monitoring challenges

# 2.3.1 Background challenge

The challenge of determining the background concentration of CO<sub>2</sub> upwind of the city is connected to wind patterns, natural fluxes and anthropogenic emissions. Higher wind speeds result in larger influence regions ("footprints") and reduce the impact of strong local sources within the background region. This leads to more spatially representative background observations and is one reason for excluding low-wind-speed observations from further analyses, such as in the inverse modelling studies over Paris by Bréon et al. (2015) (>2 m s<sup>-1</sup>) and Lian et al. (2023) (>3 m s<sup>-1</sup>).

Wind direction is also relevant for obtaining spatially representative observations, as fluxes in the dominant wind direction contribute most to the signal. Even at higher wind speeds, significant influence from large point sources or an especially active biosphere can still occur. To account for this, the emission intensity and the share of cropland surrounding the city are considered, with extra weight given to the area in the dominant wind direction. Cropland is singled out because of the added difficulty in correctly representing associated fluxes, which are influenced by crop cycles and management practices. A final consideration is wind direction. When it predominantly comes from one direction, fewer background towers are needed to

provide suitable upwind values for most observations. This makes the city less challenging to monitor in terms of the background challenge.

# 2.3.2 Biospheric challenge

The carbon landscape of cities includes the natural exchange of CO<sub>2</sub> through soils and the biosphere. Understanding the spatial and temporal distributions of these exchanges is necessary to isolate the contribution of anthropogenic emissions from observed CO<sub>2</sub>. To estimate how challenging this might be, the natural and anthropogenic fluxes as well as land cover are considered. Whereas models can be used to estimate the signal from the biosphere, these estimates are associated with large uncertainties—especially in urban environments. Therefore, strong biospheric activity in the city is expected to add to this challenge. Further adding to the challenge is when the signal from the biosphere is large in comparison to that from the anthropogenic emissions, as the signal-to-noise ratio then decreases (e.g. Sargent et al., 2018; Winbourne et al., 2022). If the city-wide biogenic signal originates from a coherent area, such as a large park, the challenge is reduced because partitioning the observations becomes easier. This is mainly relevant when observing direct fluxes in a city, as the influence areas ("footprints") are much smaller compared to influence areas of concentration measurements (Kljun et al., 2015).

# 2.3.3 Modelling challenge

- For the challenge of modelling CO<sub>2</sub> exchange within the city, both anthropogenic emissions and the city's natural and urban topography are considered. Especially point sources add complexity to this challenge. They emit large quantities of CO<sub>2</sub> from high stacks and require high-resolution spatiotemporal data and models. Maier et al. (2022) demonstrated that resolving emissions from stacks, as opposed to ground-level sources, is important even in regional-scale modelling within 50 km of the emission source. Furthermore, large shares of emissions from point sources can obscure more distributed sources, making these harder to monitor. The distribution of remaining non-point source emissions is also relevant to the modelling challenge. Spatially concentrated emissions are generally easier to monitor because they limit the spatial scope of the monitoring network and increase the likelihood of detecting large emission signals. Larger emission signals enhance the signal-to-noise ratio and thereby delay the time at which a monitoring network no longer can detect the—hopefully—decreasing emissions (Albarus et al., 2024).
- When it comes to the natural and urban topography, high shares of flat, uniform topography and low buildings reduce airflow complexity. This, in turn, makes it easier to model atmospheric transport.

# 2.3.4 Observational challenge

The metrics in this challenge relate to specific observational methods that are not covered in the other challenges: using radiocarbon to distinguish between fossil fuel and biogenic components and using satellites to make XCO<sub>2</sub> observations. As mentioned in the introduction, there are additional observational methods, and these may be preferred especially if the two

discussed here prove challenging (World Meteorological Organization, 2025). A well-established issue in using radiocarbon to infer ffCO<sub>2</sub> is radiocarbon emissions from nuclear facilities. "Potential nuclear masking" refers to the underestimation in ffCO<sub>2</sub> signal the nuclear contribution is modelled to cause if ignored. It is called "potential" because it can be corrected for, but large uncertainties in the correction arise from the quality of emission data and uncertain transport modelling. Hence, the challenge increases with the magnitude of the potential nuclear masking. A preferred practice is to avoid sampling when the nuclear contribution is expected to be significant. However, this can lead to sampling bias which is estimated for the "nuclear sample selection bias" metric. Ideally, the ffCO<sub>2</sub> signal should be of similar magnitude in both avoided and collected samples. A greater difference means a greater sampling bias and adds to the observational challenge.

The metrics related to making observations using satellites is based on cloud cover. Summer and winter are considered as separate metrics as there can be large differences between the seasons. Higher shares of cloud cover will limit the samples from future satellite missions, thereby adding to the observational challenge.

# 2.4 Integration and analysis of city characteristics

The collected metrics for the 96 cities are further analysed using statistical methods. These methods include the calculation of challenge scores and similarity matrices for the four individual challenges and an overall challenge score. In turn, the similarity matrices facilitate similarity searches and cluster analyses.

To prepare the collection to be combined, the selected metrics are transformed using a min-max normalisation between the 10-90 percentile. All cities in the 10th percentile are assigned the value of zero, and those beyond the 90th percentile are assigned the value of one. The remaining cities are scaled linearly between zero and one. The 10-90 percentile range is used to focus the analyses on the typical range of values across cities. Without normalisation, a large outlier could receive a value of one, while all other cities would get values close to zero. Even after scaling, the metric "Emission intensity buffer" still shows a strong disparity: the city in the 90th percentile has a value 17 times greater than that in the 10th percentile (see Table 2). The effect is evident in Fig. 3b, where the distribution is strongly skewed.

In most cases, a higher value of a metric can be interpreted as more challenging to monitor. However, the opposite is true for the metrics "Fraction of time wind from the dominant wind direction (limited to wind speed >2 m s<sup>-1</sup>)", "Fraction of time wind speed >2 m s<sup>-1</sup>, and "Share of flat areas". Therefore, the scaled values are inverted to ensure all metrics are interpreted in the same way before being combined into challenge scores.

# 2.4.1 Weights

To reflect that some metrics are expected to contribute more to the challenges than others, they are weighted as specified in Table 1. For the overall challenge, the scores of each of the four identified challenges are weighted equally. The individual weights within a challenge are assigned based on our literature review (as presented in Sect. 1) and experience in the field. A sensitivity analysis was performed to assess how the overall challenge score changed under different weighting schemes. Naturally, cities whose metrics almost exclusively indicate that they are either relatively hard or relatively easy to monitor will show more robust challenge scores. Once more data becomes available to link the different metrics to how well an area can be monitored, weights may be assigned in a more quantitative way. However, it should always be possible to adjust choices to accommodate the different needs of stakeholders and to recognize the value of local expert knowledge.

### 2.4.2 Challenge scores

The scaled and weighted characteristics are combined to create individual and overall challenge scores which range between zero and one, or 0 and 100%, for minimum and maximum relative challenge. The minimum and maximum values can be achieved if a city consistently falls within the bottom 10th or top 90th percentile for all metrics.

# 2.4.3 Similarity matrices

In addition to creating challenge scores, the scaled and weighted characteristics are used to generate similarity matrices based on Euclidean distances. The Euclidean distance, D, between two cities x and y is calculated as follows:

 $D(x,y) = \sqrt{\sum_{i=1}^{n} (x_i - y_i)^2}$  (2)

Where  $x_i$  and  $y_i$  represent the *i*th scaled and weighted metric scores of for cities x and y respectively. Distances are computed for all city pairs, resulting in 96×96 matrices for each of the individual challenges as well as for the overall challenge.

Similarity matrices created using Euclidean distances are suitable for further analyses, including hierarchical clustering, discussed next.

# 2.4.4 Dendrogram cluster analysis

Based on the similarity matrix for the overall challenge (see Sect 2.4.3), a dendrogram is constructed. A dendrogram is a tree-like diagram that visually represents hierarchical clusters. It starts with each city represented as an individual branch. The branches are incrementally merged according to their similarity. There are different strategies for this merging, and we use a strategy called "Ward's method" where the total within-cluster sum  $\Delta SS$  of squared Euclidean distances is minimized:

 $\Delta SS(C_i, C_j) = \frac{|c_i| \cdot |c_j|}{|c_i| + |c_j|} \cdot D(\underline{x}_i, \underline{x}_j)^2$ (3)

Where  $\Delta SS$  is calculated for all possible combinations of two clusters,  $C_i$  and  $C_j$ , that can be merged.  $|C_i|$  and  $|C_j|$  represent the number of cities within each cluster.  $\underline{x}_i$  and  $\underline{x}_j$  are the centroids of these clusters. The Euclidean distances between the centroids are calculated using Eq. 2.

As clusters are merged, the dendrogram moves towards forming a single branch (see Fig. 4). The later that two branches are merged, the more dissimilar the cities in the two branches are. Before merging, the branches can be viewed as individual clusters. Visual inspection of the dendrogram tree reveals a set of meaningful clusters, discussed further in the result section.

### 3 Results

The results begin with a section that highlights some of the individual characteristics of the cities and exemplifies what several of the input spatial data layers look like (see Fig. 2). Next, the challenge scores estimated from the combination of metrics are presented, followed by their application in similarity searches. Finally, general similarities and dissimilarities among all cities are identified based on the cluster analysis result. There is a general focus on Paris, Munich, and Zurich as these are part of the evolving urban observation network within ICOS (https://www.icos-cp.eu/projects/icos-cities, last access: October 2024). Similarity searches are employed to identify the potential for knowledge exchange between cities that face similar challenges to those within the network. Finally, the cluster analysis is used to identify cities that are dissimilar to those already in the network. These cities are argued as good candidates for additions to the ICOS Cities network. More details about other specific cities can be found in the resources published along with this study (see Sect. 5).

# 3.1 General characteristics

The 96 selected cities exhibit a wide range of values across the different characteristics with 90th percentile values that are often several times higher than the 10th percentile values (see Table 2). The 10th-to-90th percentile span is most extreme for metrics that include emissions from point sources. While many cities have none, some have large emitters that account for almost all the emissions in the city. Furthermore, the non-Gaussian distribution of large emission sources contributes to high variability of emission intensity in the surroundings of the cities and partly explains major differences in the ratio between NEE and anthropogenic CO<sub>2</sub>. Nuclear facilities are also unevenly distributed, with particularly large amounts of radiocarbon emitted from La Hague, located on the coast of Normandy, France. This creates significant "nuclear masking potential" in a handful of cities that are close and results in a mean value that is as large as the 90th percentile. There are also significant differences in the sampling bias introduced by adopting the strategy of discarding samples with large nuclear contributions.

The large differences between cities lead to very different challenges when it comes to emission monitoring, confirming the primary motivation for this study.

| Metric                                                                                          | Unit                                 | Mean | 10th percentile | 90th percentile | Std.  |
|-------------------------------------------------------------------------------------------------|--------------------------------------|------|-----------------|-----------------|-------|
| Fraction of time wind speed >2 m s <sup>-1</sup>                                                | %                                    | 84   | 73              | 93              | 10    |
| Fraction of time wind from dominant                                                             | %                                    | 26   | 21              | 32              | 5     |
| wind direction (limited to wind speed                                                           |                                      |      |                 |                 |       |
| >2 m s <sup>-1</sup> )                                                                          |                                      |      |                 |                 |       |
| Emission intensity buffer                                                                       | tCO <sub>2</sub> km <sup>-2</sup>    | 5264 | 750             | 14621           | 6899  |
| Emission intensity buffer dominant wind direction (limited to wind speed >2 m s <sup>-1</sup> ) | tCO <sub>2</sub> km <sup>-2</sup>    | 5342 | 368             | 14994           | 12051 |
| Share of point source emission                                                                  | %                                    | 29   | 0               | 76              | 28    |
| Non-point-source emission spatial                                                               | %                                    | 19   | 12              | 25              | 5     |
| aggregation                                                                                     |                                      |      |                 |                 |       |
| Vegetation heterogeneity                                                                        | %                                    | 24   | 14              | 34              | 7     |
| Share cropland buffer                                                                           | %                                    | 30   | 12              | 51              | 14    |
| Share cropland buffer dominant wind direction (limited to wind speed >2 m s <sup>-1</sup> )     | %                                    | 30   | 5               | 57              | 21    |
| NEE relative to emissions                                                                       | %                                    | 25   | 8               | 44              | 22    |
| Average NEE                                                                                     | μmol m <sup>-2</sup> s <sup>-1</sup> | 0.60 | 0.39            | 0.81            | 0.18  |
| Average building height                                                                         | m                                    | 7.2  | 5.5             | 8.9             | 1.2   |
| Share flat areas                                                                                | %                                    | 44   | 12              | 71              | 22    |
| Topographic heterogeneity                                                                       | m                                    | 2.6  | 1.1             | 5.1             | 1.8   |
| Nuclear masking potential                                                                       | %                                    | 20   | 4.6             | 19.7            | 108   |
| Nuclear sample selection bias                                                                   | %                                    | 19   | 3.9             | 38              | 12    |
| Share days >30% clouds summer                                                                   | %                                    | 74   | 68              | 82              | 5.9   |
| Share days >30% clouds winter                                                                   | %                                    | 88   | 81              | 95              | 5.6   |

Table 2: Averages, standard deviations, 10th- and 90th-percentile values for the 18 metrics based on the 96 analysed cities.

Figure 2: Four of the input data layers subset for Zurich, showing (a) natural topography, (b) land cover, (c) biosphere net ecosystem exchange (NEE), and (d) total ffCO<sub>2</sub>. The largest green point in the CO<sub>2</sub> emission map (d) represents Zurich's airport and falls just outside the city border. The biogenic flux map (c) is based on an average from wintertime afternoons in 2018 (see Sect. 2.2.3).

Figure 3 shows all 18 metrics for the ICOS Cities pilot cities, Munich, Zurich, and Paris. Paris stands out among the other cities for its relatively low citywide NEE relative to its large ffCO<sub>2</sub> emissions, which makes emission monitoring easier. However, the NEE in Paris is associated with fragmented vegetation, as indicated by the high vegetation heterogeneity metric. One implication is that signals from emissions are mixed with signals from biogenic activity, making it difficult to isolate them. Another factor indicating that Paris is relatively challenging to monitor is its average building height of 8.9 meters, which falls in the 90th percentile. This complex urban topography complicates the transport modelling.

Munich and Zurich both have strong dominant wind directions. This is advantageous for representing the inflow boundary conditions with a limited network of tall tower stations measuring concentrations. However, compared to the other cities the wind speed is quite frequently below 2 m s<sup>-1</sup>. This rather adds to the challenge, as upwind observations are less likely to be spatially representative during periods of low wind-speed. Both cities have low shares of emissions from point sources and are not expected to have a major problem with nuclear contribution in potential radiocarbon samples. Figure 2d shows the point sources in Zurich, but we note that the largest point source—Zurich's airport—lies just outside the city boundaries and is therefore not included in the metric "share of point source emission". Airports cannot be represented with take-off and landing information in the TNO emission inventory and are therefore represented by point sources which keep their exact location.

All three cities differ significantly when it comes to natural topography; Zurich stands out with only 6% flat areas and a high topographic variability, placing it in the 90th percentile for both these metrics (see Fig. 2a). As in Paris, with its complex urban topography, this will make modelling in Zurich particularly challenging. Out of the three cities, Munich has the most advantageous natural and urban topography for monitoring ffCO<sub>2</sub> emissions.

Figure 3: (a) The 18 metrics listed along the y-axis are linearly scaled between the values of the city at the 10th percentile and the city at the 90th percentile, out of the 96 cities (see Table 2). They are organized along the y-axis according to their association with the four discussed challenges. Higher values indicate greater challenges to monitor CO<sub>2</sub> emissions. (b) Density plot showing where most cities fall in the linear scaling between the 10th and 90th percentile.

# 3.2 Challenge scores

The overall challenge scores (see Sect. 2.4.2) range from 30% for Leiden, the Netherlands—indicating a relatively low challenge—to 59% for Rouen, France (see Table 3). The biogenic and modelling challenges contribute the most to these scores for the two cities, respectively. No clear spatial patterns are observed in which challenge dominates across nearby cities, except around the Ruhr area in western Germany (see Fig. 1). Here, many cities can expect challenges to determine background concentrations. A main driver is that many of these cities are close to each other, which results in high emission intensity in their surroundings, thereby increasing the background challenge.

Among the three target cities, Munich has a low overall challenge score (34%), close to that of Leiden. Compared to other cities, the scores associated with the biogenic challenge and modelling challenges are particularly low (see Table 3). Like for Paris, the ratio between NEE and ffCO<sub>2</sub> emission is small, and for Munich the average NEE is also relatively low placing the city in the 10th percentile of the biospheric challenge (see Fig. 3). While the overall score of Zurich is similar to that of Munich, there are differences between individual challenges. In particular, the modelling challenge stands out due to the Zurich's complex urban and natural topography.

Paris has the highest overall score of the three and stands out for its high score in the challenge of determining background concentrations. Paris is also in the third quartile when it comes to the observational and modelling challenges. Contributing factors include a high concentration of emissions from point sources and tall buildings, as well as high cloud cover, especially in the summer. The cloud cover likely reduces the number of useful satellite observations. The influence of nuclear emissions is the highest among the three pilot cities but remains relatively low compared to all 96 cities considered.

| City          | Overall |          | Background |          | Biogenic |          | Modelling |          | Observational |          |
|---------------|---------|----------|------------|----------|----------|----------|-----------|----------|---------------|----------|
|               | %       | Q and R* | %          | Q and R* | %        | Q and R* | %         | Q and R* | %             | Q and R* |
| Munich, DE    | 34      | Q1 (6)   | 31         | Q2 (25)  | 30       | Q1 (9)   | 26        | Q1 (17)  | 50            | Q3 (65)  |
| Zurich, CH    | 35      | Q1 (9)   | 20         | Q1 (5)   | 34       | Q1 (19)  | 54        | Q3 (70)  | 32            | Q2 (26)  |
| Paris, FR     | 45      | Q3 (65)  | 45         | Q3 (61)  | 38       | Q2 (29)  | 50        | Q3 (57)  | 48            | Q3 (61)  |
| Leiden, NL    | 30      | Q1 (1)   | 32         | Q2 (30)  | 33       | Q1 (17)  | 30        | Q2 (26)  | 24            | Q1 (16)  |
| Rouen, FR     | 59      | Q4 (96)  | 38         | Q2 (41)  | 41       | Q2 (35)  | 80        | Q4 (96)  | 76            | Q4 (92)  |
| Kassel, DE    | 40      | Q2 (34)  | 14         | Q1 (1)   | 50       | Q3 (66)  | 56        | Q4 (76)  | 41            | Q2 (41)  |
| Groningen, NL | 45      | Q3 (49)  | 70         | Q4 (96)  | 66       | Q4 (85)  | 9         | Q1 (5)   | 38            | Q2 (28)  |
| Rennes, FR    | 40      | Q2 (29)  | 42         | Q3 (58)  | 18       | Q1 (1)   | 40        | Q2 (40)  | 57            | Q4 (80)  |
| Gliwice, PL   | 42      | Q2 (45)  | 27         | Q1 (17)  | 76       | Q4 (96)  | 18        | Q1 (6)   | 47            | Q3 (57)  |
| Almere, NL    | 32      | Q1 (3)   | 39         | Q2 (46)  | 49       | Q3 (61)  | 1         | Q1 (1)   | 39            | Q2 (38)  |

| Rouen, FR      | 59 | Q4 (96) | 38 | Q2 (41) | 41 | Q2 (35) | 80 | Q4 (96) | 76  | Q4 (92) |
|----------------|----|---------|----|---------|----|---------|----|---------|-----|---------|
| Düsseldorf, DE | 39 | Q2 (28) | 64 | Q4 (94) | 39 | Q2 (31) | 51 | Q3 (62) | 0.3 | Q1 (1)  |
| Dijon, FR      | 55 | Q4 (95) | 36 | Q2 (36) | 38 | Q2 (28) | 53 | Q3 (69) | 93  | Q4 (96) |

<sup>\* &</sup>quot;Q & R" stands for Quartile and Rank.

Table 3: Challenge scores for Paris, Munich, and Zurich along with the cities with the highest and lowest scores overall, and for each of the four challenges. The higher the score, the greater the anticipated challenge.

### 3.3 Similarity searches

Similarity matrices are used to quantify the potential to transfer the CO<sub>2</sub> monitoring experience gained from the three ICOS pilot cities, exemplified here with Munich. In terms of similarities relevant to the background challenge, Linz (Austria), Mulhouse (France), and Augsburg (Germany) are most like Munich (see Table 4). These are cities where, as in Munich, this challenge is relatively low (see Table 3). In practice, this could mean that only a few background towers are needed in the outskirts of the cities to obtain representative boundary conditions for most situations. The biogenic challenge in Munich is also low, as similar cities include Brussels (Belgium), Nantes, and Lille (France). It will not be as difficult to separate the anthropogenic signal in these cities as it is in cities at the opposite end of the spectrum from Munich. Cities such as Bratislava (Slovakia) and Erfurt and Hagen (Germany) are listed as the most dissimilar to Munich in this aspect (see Table 4).

Out of the 96 cities, Nuremberg is overall the city most like Munich, while the corresponding cities for Zurich and Paris are the German cities Kassel and Berlin. Their monitoring strategies could look similar, but to overcome individual challenges it may still be useful to consider similarities in terms of the specific challenges. In terms of the background challenge, Karlsruhe (Germany) is most similar to Zurich, and Charleroi (Belgium) to Paris. Charleroi is also most like Paris regarding the biogenic challenges, and for Zurich, the corresponding city is Brussels. Tables listing the top five most similar cities to each of the 96 cities across the different challenges are provided in the mapbooks (see Sect. 5).

| Overall (%)        | Background (%)    | Biogenic (%)        | Modelling (%)               | Observational (%)   |
|--------------------|-------------------|---------------------|-----------------------------|---------------------|
|                    |                   | Most similar        |                             |                     |
| Nuremberg, DE (92) | Linz, AT (98)     | Brussels, BE (100)  | Tilburg, NL (97)            | Graz, AT (97)       |
| Vienna, AT (92)    | Mulhouse, FR (98) | Nantes, FR (100)    | Angers, FR (96)             | Vienna, AT (97)     |
| Augsburg, DE (91)  | Augsburg, DE (96) | Lille, FR (100)     | Orléans, FR (96)            | Bratislava, SK (95) |
| Hanover, DE (91)   | Ostrava, CZ (95)  | The Hague, NL (100) | Lens, FR (96)               | Gliwice, PL (94)    |
| Paris, FR (91)     | Zurich, CH (93)   | Antwerp, NL (100)   | Mönchengladbach,<br>DE (95) | Wrocław, PL (92)    |
|                    |                   | Most dissimilar     |                             |                     |

| Haarlemmermeer, NL | Haarlemmermeer,    |                      |                    |                 |
|--------------------|--------------------|----------------------|--------------------|-----------------|
| (80)               | NL (62)            | Bratislava, SK (42)  | Antwerp, NL (66)   | Nates, FR (56)  |
|                    |                    |                      |                    |                 |
| Gdynia, PL (80)    | Groningen, NL (63) | Erfurt, DE (42)      | Karlsruhe, DE (67) | Rennes, FR (56) |
|                    |                    |                      | Gelsenkirchen, DE  |                 |
| Odense, DE (81)    | Cologne, DE (64)   | Hagen, DE (42)       | (67)               | Lens, FR (57)   |
|                    |                    |                      |                    |                 |
| Groningen, NL (81) | The Hague, NL (65) | Münster, DE (42)     | Linz, AT (67)      | Angers, FR (59) |
|                    |                    |                      |                    |                 |
| Alkmaar, NL (81)   | Rotterdam, NL (65) | Saarbrücken, DE (42) | Mannheim, DE (68)  | Reims, FR (59)  |

Table 4: Similarity to Munich in terms of the four individual challenges, as well as overall similarity when the four challenges are combined ("overall challenge"). A higher value indicates greater similarity.

### 3.4 Cluster analysis

As a complement to the similarity searches, the results from a dendrogram cluster analysis shows the overall structure of similarities and dissimilarities across all 18 metrics (see Fig. 4). The matching of cities with the ICOS Cities pilot cities, as exemplified in Sect. 3.3, could be improved for many of the 96 cities if more clusters were represented by pilot cities. Hence, the dendrogram can be used to guide future network expansion. Munich and Paris both fall into the same cluster, C1, whereas Zurich is in cluster C4 (see Fig. 4). The hierarchical structure of the dendrogram shows that cities in cluster C3, followed by those in cluster C5, are the furthest away in the cluster space from the already represented clusters. A prominent city in cluster C3 is Copenhagen, Denmark. Its characteristic signature (see CMC-CITYMAP; Sect. 5) indicates that Copenhagen is expected to face a greater biogenic challenge compared to the pilot cities. Using complementary observations of correlated trace gases or isotopes to separate the ffCO<sub>2</sub> signal will be especially important in similar cities. However, the use of  $\Delta^{14}$ C would come with the additional uncertainty of accounting for nuclear emissions which have a significant influence in Copenhagen. This aspect of the city adds to its observational challenge. Both the background and the modelling challenges are relatively minor: The main challenges stem from the lack of a dominant wind direction and a high average building height, though the latter is still lower than in Zurich and Paris.

Bratislava, Slovakia, is a good candidate from cluster C5 and faces an even higher biogenic challenge than Copenhagen. However, the vegetation is relatively clustered in space which makes the signal less mixed. Bratislava also stands out for its high share of cropland surrounding the city, which complicates the determination of representative background levels of CO<sub>2</sub>. A solution there could be to deploy more background sites to capture the potential heterogeneity of cropland fluxes. Cities in the final cluster, C2, are located closer in cluster space to those that already include pilot cities (see Fig. 4) but could be prioritized next.

Figure 4: Dendrogram based on the similarity matrix created from all 18 metrics included in the overall challenge. The different colours represent five distinct clusters formed by drawing a horizontal line at the desired separation between the dendrogram branches. Paris (C1), Munich (C1), and Zurich (C4) are highlighted on the x-axis.

### 4 Discussion

The metric scores in this study are derived from the relative distribution of the selected cities which are all from a subregion in western Europe (see Fig. 1). If cities from a broader geographic area, spanning different climate zones, were included, the range of values would likely change. For example, we would expect a wider range in the cloud cover metric, as some regions experience consistently cloudy conditions for part of the year. Large point sources in additional cities could further increase the already high 90th percentile values in related metrics. In our dataset, 13 out of the 96 cities account for 75% of the point source emissions, resulting in the skewed distribution seen in Fig. 3b. A city like Paris—with 19% of emissions from point sources—receives a score of only 0.25 out of one, where one indicates the highest level of challenge.

Another important aspect affecting our results is how our cities are defined geographically. Our city borders are based on the OECD definition of a city (Dijkstra et al., 2019), but these still rely on local administrative boundaries provided by the countries. Albarus et al. (2023) observe that the drawing of administrative boundaries sometimes results in cities being separated from large portions of emissions in their immediate surroundings. Other times, the boundaries may include extended areas of nonurban land cover. The former scenario places greater demands on CO<sub>2</sub> emission monitoring to distinguish between emissions within and outside the borders (Albarus et al., 2023). This issue is partly addressed in our study, as adjacent administrative units with high population densities form a single city (Dijkstra et al., 2019). However, significant nearby

emission sources may still be excluded, as in Zurich, where the airport falls just outside the city boundaries (see Fig. 2d). The effect of this is mitigated by the fact that the airport still contributes to the city's challenge scores through its inclusion in the 20-kilometer buffer area (see Sect. 2.1). One option could be to consider emission intensity, rather than population, as a criterion for merging local administrative units in the OECD approach. This would preserve the advantage of integrating readily available statistics from local administrative units in future analyses. Another alternative could be to define city boundaries entirely based on the highest-resolution emission data available, creating so-called "carbon cities". This approach would likely reduce the inclusion of large nonurban areas on the outskirts of cities, which particularly affects our urban vegetation-related metrics.

Our selection of metrics and how they are synthesised into four challenges are motivated by our literature review (as presented in Sect. 1) and experience in the field. Some of the studies present results that can be discussed in the context of our findings. Previous studies in Paris shed light on what we refer to as the "background challenge", where Paris scores in the 3<sup>rd</sup> quartile. The relatively high score for Paris aligns with the findings of Sargent et al. (2018). They warned that boundary conditions can be particularly complex for continental cities due to long- and medium-range transport from both distant urban areas and biogenic sources. Lian et al. (2021) indeed found especially large discrepancies between different modelled boundary conditions when air was coming from continental Europe—up to 5 ppm between two products. This is significant, as the CO<sub>2</sub> gradients between urban and suburban "background" towers in Paris were found to be 5-10 ppm in the summer and 20-30 ppm in the winter (Lian et al., 2023). In cities or regions with lower emission intensities than Paris, a bias in the boundary conditions would be even more impactful. For example, Lauvaux et al. (2012) found that a 0.55 ppm bias in the boundary condition resulted in a substantial impact on the posterior annual CO<sub>2</sub> flux for Iowa and the surrounding states.

Best practices proposed to mitigate the "background challenge" include using observations to find upwind-downwind gradients for inversions (e.g., Bréon et al., 2015; Staufer et al., 2016), or to constrain the modelled boundary conditions with observations (e.g., Sargent et al., 2018). Our metrics associated with the challenge offer an estimate for how spatially representative the observations may be by considering fluxes nearby the cities. Our consideration of wind speed and direction also ties to how many useful observations would be available for the different practices aimed at limiting the bias from boundary conditions. These factors greatly reduced the number of samples that could be used in the inversion over Paris by Bréon et al. (2015). At the time, the background concentration was sampled from only two towers, and the wind speed threshold, like ours, was 2 m s<sup>-1</sup>. Compared with the other cities, Paris is among the most favourable in terms of wind speed but exhibits one of the most variable wind directions. Today, more background towers are available, and cities with similar characteristics could also benefit from deploying a larger number of background sites from the outset.

Regarding the "biogenic challenge", Lian et al. (2023) highlighted their poorly resolved and non-optimised biogenic fluxes as a key area for improvement in future studies in Paris. It was pointed out as a likely contribution to the 20% increase in their

optimised ffCO<sub>2</sub> estimates compared to the emission inventory used as prior (April-June). Different borders for Paris (Lian et al., 2023; Fig. 1) are just one of the reasons we cannot directly compare our results, but based on our analysis the significance of the biosphere is not surprising: even in winter afternoons the modelled net influence of the biosphere is 8% compared to the ffCO<sub>2</sub> emissions. On summer afternoons the NEE is more than twice the magnitude of the anthropogenic emissions. If we instead consider borders roughly bounded by Le Bourget Airport in the north and Paris-Orly in the south, the corresponding values are 0.9% and 11%, which are more in line with the findings and adjustments to the ffCO<sub>2</sub> emissions in Lian et al. (2023). It is also consistent with the work by Albarus et al. (2024), who observed much lower signal-to-noise ratios further away from the Paris city centre. However, even given the borders extending further into the area with a lower ffCO<sub>2</sub> signal-to-noise ratio, Paris has a low biogenic challenge score compared to most of our cities (2<sup>nd</sup> quartile). Hence, even cities with low scores likely require the use of well-calibrated biospheric models, preferably optimised with complementary direct flux measurements and observations of correlated tracers and/or isotopes. This is quite likely preferable to the strategy of using observations only in the dormant season (e.g. Lauvaux et al., 2016), as this comes with the additional uncertainties of using temporal profiles to scale the results to the rest of the year (Super et al., 2020; Super et al., 2021).

For the "modelling challenge" most of the metrics are related to the complexity of natural and urban topography, which puts high demands on models to accurately resolve the airflow. This is the main driver for Zurich's challenge score (3<sup>rd</sup> quartile). However, the study by Berchet et al. (2017) conducted in Zurich shows good performance of their model, which they found to fulfil the requirements for air pollution modelling at most of the tested sites. Although the requirement for modelling CO<sub>2</sub> is higher, this is promising for cities' abilities to overcome this challenge. Hence, cities with scores similar to that in Zurich could benefit from adopting such a model. The challenge for models to accurately represent nearby point source emissions is also well-established (e.g. Gaudet et al., 2017; Maier et al., 2022; Brunner et al., 2019). This challenge is compounded by large emission quantities stemming from these sources, which generally do not have point-source-specific temporal profiles. While hourly emissions are sometimes available, such as for many power plants throughout Europe, most models currently cannot include them.

The "application-specific observational challenge" currently combines metrics related to how well-suited cities are for making satellite and radiocarbon observations. They can be evaluated independently in Fig. 3, and stakeholders interested in specific cities can consider the two observational methods separately in the mapbooks (Storm et al., 2025b, <a href="https://doi.org/10.18160/Z66D-05JT">https://doi.org/10.18160/Z66D-05JT</a>). The satellite section currently only includes cloud cover, as this is a crucial factor, affecting the number of expected samples (e.g. Kuhlmann et al., 2019). However, the relevance of satellite observations to our study is debatable, as only a limited number of cities (15) had emission quantities greater than 7.33 MtCO<sub>2</sub> yr<sup>-1</sup> in 2018—the threshold suggested by Wang et al. (2020) as appropriate for monitoring emissions from space with the CO2M instrument.

For the use of <sup>14</sup>CO<sub>2</sub> observations, the observational challenge is linked to how much contribution is expected from emissions from nuclear facilities. As in previous studies (e.g., Maier et al., 2023), we used a flat annual emission rate to simulate this, but improving the resolution of this emission data is a priority at the ICOS Radiocarbon Laboratory. For example, knowing the timing of emissions from La Hague, France, would significantly enhance the feasibility of using radiocarbon in many cities beyond those closest to it. In 2021, La Hague accounted for 39% of the <sup>14</sup>C in CO<sub>2</sub> emissions from European nuclear facilities (Storm et al., 2024b), with large quantities released during short periods. Excluding La Hague's emissions from our analyses, thereby simulating conditions between major emission events, reduces the nuclear masking potential's 10th to 90th percentile range from 5-20% to 3-11%. This highlights how our findings can guide and motivate future efforts and underscores the importance of updating our analyses as new data becomes available to the community. In addition, the nuclear challenge also depends on the <sup>14</sup>CO<sub>2</sub> sampling strategy to be established within the city: When coordinated upwind and downwind sampling is employed, it can be assumed that most of the nuclear contribution will be captured in the up-and-downwind samples and is thus intrinsically corrected.

Our focus has been on placing our results within the context of existing urban CO<sub>2</sub> monitoring studies, with particular attention to our three pilot cities. While it was not feasible nor possible to evaluate each individual metric and its true relevance to the challenges, our framework offers a foundation for future discussion and refinement as the research field progresses. Within ICOS Cities, it can support the project vision of developing "blueprints" for monitoring emissions in European cities. We recommend a modular approach for this, enabling cities to match with and adopt strategies from the pilot city that are most similar in ways relevant to the specific challenges. This approach is comparable to that of the "Twinning Learning Program", part of the European Union's mission "100 Climate-Neutral and Smart Cities by 2030", where cities are paired based on shared barriers to achieving climate neutrality. From a pan-European monitoring strategy perspective, it is important to develop blueprints for strategies that are effective across the diverse characteristic signatures found in Europe. To support this, we identified Bratislava and Copenhagen as cities that are among the most distinct from the three cities currently in the ICOS Cities network, making them strong potential candidates for inclusion into the network. This assessment considered all metrics in combination. A modular approach could also be applied here. Cities with high scores in the "biogenic challenge"—which is low for the three pilot cities relative to the others—would then be highlighted as especially suitable candidates. Bratislava would again be among the recommended cities. All in all, there are numerous ways our framework can be used to create analyses like those presented in this study. Adjustments could range from minor changes to the weights of the 18 metrics to entirely different analyses based on a new selection of metrics that are readily available for our cities but not used here.

### 5 Data availability

For the datasets used to derive the metrics in this paper, we refer to the cited references. The resulting collection of 18 metrics, along with several metrics excluded from the study, is published along with the notebook tool (Storm et al., 2025a,

https://doi.org/10.18160/P8SV-B99F). The tool, provided as a Jupyter Notebook with accompanied Python files, can also be run directly in the Interactive Computing Environment offered by the ICOS Carbon Portal. Individual PDFs, referred to as "mapbooks", contain maps and analysis results for all cities. These are published as a collection and can be downloaded for the individual cities (Storm et al., 2025b, https://doi.org/10.18160/Z66D-05JT).

### **6 Conclusions**

- This study presents a methodology to understand and quantify the differences between cities and what these differences mean from a CO<sub>2</sub> emission monitoring perspective. We analyse 96 cities in western Europe using 18 defined metrics, linking these metrics to four key CO<sub>2</sub> monitoring challenges. Next, the challenges are quantified to provide insights into the evolving network of urban observatories in Europe, with a focus on the ICOS Cities pilot cities: Paris, Munich, and Zurich. Their relationships to the other 93 cities are quantified to illustrate: 1) which monitoring challenges may be most significant, 2) which cities are similar and could benefit from knowledge exchange, and 3) which cities are dissimilar and may serve as candidate cities if there is funding to expand the ICOS Cities network.
- Overall, our results suggest that Zurich and Munich are relatively easy to monitor, with Zurich facing the greatest challenge in the "modelling challenge" and Munich in the "application-specific observational challenge". Paris scores similarly to Zurich in the modelling challenge but has high scores in the other challenges except for the "biogenic challenge". Cities similar to Munich are identified across the different challenges, suggesting, for instance, that monitoring strategies used to address the background challenge in Munich may also be effective in for example Linz (Austria). Paris, Munich, and Zurich fall into two out of five clusters when considering all 18 metrics. Copenhagen and Bratislava are highlighted as prominent cities in clusters that are currently not represented by the ICOS Cities network. These could be interesting candidates if an extension to the pilot network is considered.
- We have only highlighted a few examples from the results, which represent just a subset of the potential analyses that can be drawn from the framework we have developed. We refer to Sect. 5 for how to access results for specific cities of interest or to conduct new analyses based on a different set of characteristics.
- As the field of urban emission monitoring continues to evolve, we anticipate ongoing developments that will both help mitigate current challenges and enhance our suggested methods for analysing them. Regarding the mitigation of challenges, we have highlighted, among other examples, the need for improved urban-specific biogenic models and transport models capable of resolving airflow in urban environments. The research continues to advance, and datasets with higher accuracy than the one used in this study are already available for individual cities. As use of these models and data becomes more widely adopted, they can be applied in analyses similar to ours. With more training data, we may start to better understand the relationships

- between different city characteristics and their influence on the ease of monitoring emissions. One possible approach is to use
- machine learning, correlating model—data differences with these characteristics.

# 703 7 Author contribution

- IS, UK, AV, and WP designed the study. SH contributed to the design of the nuclear contribution metrics (Sect. 2.2.5). ISU
- and TG provided anthropogenic and biogenic CO<sub>2</sub> fluxes, respectively, along with insights regarding their use. IS developed
- the Jupyter Notebook tool and mapbooks together with CD'O and UK. IS prepared the manuscript, with contributions from
- all co-authors.

717

725

# 8 Competing interests

The authors declare that they have no conflict of interest.

### 9 Financial support

- This work has been supported by the ICOS PAUL project: PAUL, Pilot Applications in Urban Landscapes Towards integrated
- city observatories for greenhouse gases (ICOS Cities), funded by the European Union's Horizon 2020 Research and Innovation
- Programme under grant agreement No 101037319.

### 714 10 Acknowledgements

- The creation of STILT footprints has received funding from the European Union's Horizon 2020 research and innovation
- programme through the ATMO-ACCESS Integrating Activity under grant agreement No 101008004.
- The biogenic fluxes used in this study were created using resources of the Deutsches Klimarechenzentrum (DKRZ) granted
- by its Scientific Steering Committee (WLA) under project bd1231. The work was financed through the CO2KI project of the
- BMBF (project number FZK50EE2212).

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

| 995  | World Meteorological Organization: IG3IS Urban Emission Observation and Monitoring Good Research Practice Guidelines                             |
|------|--------------------------------------------------------------------------------------------------------------------------------------------------|
| 996  | (GAW Report No. 314), https://doi.org/10.59327/WMO/GAW/314, 2025.                                                                                |
| 997  |                                                                                                                                                  |
| 998  | Zanaga, D., Van De Kerchove, R., Daems, D., De Keersmaecker, W., Brockmann, C., Kirches, G., Wevers, J., Cartus, O.,                             |
| 999  | Santoro, M., Fritz, S., Lesiv, M., Herold, M., Tsendbazar, NE., Xu, P., Ramoino, F., and Arino, O.: ESA WorldCover 10 m                          |
| 1000 | 2021 v200 (Version v200) [Data set], Zenodo, <a href="https://doi.org/10.5281/zenodo.7254221">https://doi.org/10.5281/zenodo.7254221</a> , 2022. |