# Peer review of "Monitoring CO2 in diverse European cities: Highlighting needs and"

_Earth System Science Data, 2025_

## Author Comment (AC1)

**RC1: 'Comment on essd-2025-63', Anonymous Referee #1, 20 May 2025**

The scope of this paper is the monitoring of CO2 emissions of European cities from the atmospheric concentrations. Indeed, the emission from the cities increases the atmospheric concentrations so that concentration measurements can be used to infer the emissions. There are a number of challenges however linked to the emission-concentration relationship that depends on the variable atmospheric transport, the heterogeneity of the emission or the other (than the city emissions) fluxes that impact the CO2 concentrations. The challenges are different among cities and this paper attempts to classify the European cities according to these challenges. They have defined a number of indicators to quantify the various challenges and make a statistical analysis based on this criteria to identify the cities that are the most suitable for CO2 monitoring experiments.

The paper is interesting and can be of interest for the growing community that attempts to estimate city emissions either from surface network measurements of from remote sensing imagery. One could certainly criticize the definition of the challenge indicators but the choice of the authors appear reasonable.

Note that some challenge apply mostly to remote sensing applications (such as the cloud cover) whereas other are more applicable to the definition of use of a surface network (such as the wind direction). All indicators are made available so that one can make its own classification.

We thank the reviewer for their assessment of our manuscript. Below, we address each of the points raised in blue. Please note that the line numbers in the reviewer's comments refer to the original version of the paper, while our responses pertain to the revised version.

Minor comments to be considered by the authors:

Line 38: Cities account for approximately: "account for" is not clear enough. Is it scope 1, scope 2 or scope 3? Only scope 1 emissions could be measured from the atmospheric concentrations

We agree with the reviewer that this needs to be clarified and have updated the text to be more specific:

Line 42: "In 2020, cities accounted for approximately 67-72% of global CO2-equivalent emissions based on consumption-based accounting (Lwasa et al., 2022)"

Consumption-based accounting not only includes Scope 1 (direct) emissions, but also Scope 2 and Scope 3 emissions. However, in this introduction, our aim is to highlight the general importance of cities.

Line 52: (such as kgCO2/vehicle),: It would make more sense to have kkCO2/km

We agree with the reviewer: it is indeed not emission per vehicle, but emission per vehicle kilometre (vkm) that is used when talking about emission factors for road transport in the context of the emission inventory we use. We have updated the text:

Line 55: "Most cities that engage in emission monitoring use "bottom-up" approaches that usually do not include direct observations: activity data (such as traffic counts) are combined with emission factors (such as kgCO2/vehicle km (vkm)), and the sophistication of its implementation varies."

The emission inventory we use reports quantities in kilograms, which is why we keep this unit.

Line 83: in Indianapolis the enhancement is only about 3 ppm. Is it on average, or max?

We have updated the text, as we agree with the reviewer that clarification was needed.

Line 88: "For example, in Indianapolis the enhancement at the downwind site was only about three ppm in October 2012 (averaged over 17-22UTC), according to Lauvaux et al. (2016)."

Line 148: "about 50 out of 365 plumes per year could". Better to say that, out of the 365 days in a year, only 50 appear suitable to observe the CO2 plume from space"

We thank the reviewer for the suggestion which we have implemented.

Line 154: "For example, in a synthetic study for Berlin, Kuhlmann et al. (2019) found that out of the 365 days in 2012, only 50 appeared suitable to observe the CO2 plume from space due to unfavourable meteorological conditions."

Line 149: Furthermore, the collected samples were higher... Not clear. What is higher? Emissions or CO2 plume?

Emissions were higher, and we see that this was not clear in the text. We have now clarified it:

Line 156: "Furthermore, the emissions during the sample collection were 18% higher than the annual total for Berlin, requiring temporal profiles to correct for this sampling bias."

Line 153. might be monitored from ». "might be monitored" lacks detail. It depends on the accuracy requirement

We agree with the reviewer and have updated the text to be more specific.

Line 159: "Wang et al. (2020) suggested that emissions from a city or a power plant larger than 7.33 MtCO2 yr-1 (2 MtC yr-1) could potentially be constrained between 8:30 and 11:30 using the CO2M instrument, which has a planned launch in 2026. The threshold corresponds to a posterior uncertainty smaller than 20% for more than 10 times within a year."

Line 449: make eddy covariance measurements ». I did not understood that this paper analyzes the possibility to make such measurements

The focus of the paper is indeed the atmospheric monitoring of CO2, and we have updated the text to provide a more relevant example to illustrate the connection between monitoring challenges and the heterogeneity of biogenic activity.

Original: "One implication is that this potentially makes it hard to find good locations to make eddy covariance measurements with limited influence of the urban biosphere."

Line 463: "One implication is that signals from emissions are mixed with signals from biogenic activity, making it difficult to isolate them."

Line 458: What emissions from airport are considered? Is it mostly that of the building or that of the plane take off? For those, the temporal profile of the emissions may be quite challenging

The building emissions are a different category, and it is rather the emissions associated with take-off and landing that are considered. They are centred on the location of the airport and treated as point-source emitters. In the paper, we state: "Airports cannot be represented with take-off and landing information in the TNO emission inventory and are therefore rather turned into point sources which keep their exact location." (line 473).

Point-source emitters, in general, are indeed challenging, partly because of their temporal profiles. Here is a relevant section of the paper:

Line 624: "The challenge for models to accurately represent nearby point source emissions is also well-established (e.g. Gaudet et al., 2017; Maier et al., 2022; Brunner et al., 2019). This challenge is compounded by large emission quantities stemming from these sources, which generally do not have point-source-specific temporal profiles. However, hourly emissions are sometimes available, such as for many power plants throughout Europe, but most models cannot include them."

There is, for example, the metric "Share of point source emissions" (see Sect. 2.2.2) as part of the modelling challenge, which airports contribute to.

---

## Author Comment (AC3)

**RC2: 'Comment on essd-2025-63', Anonymous Referee #2, 23 Aug 2025**

**1. General Comments**

This paper presents a timely framework, the "CO2 Monitoring Challenges City Mapbooks" (CMC-CITYMAP), for characterizing and classifying European cities based on the challenges they pose to urban CO2 monitoring. The manuscript is well-structured, and the methodology is generally well-explained. The authors use a systematic approach to quantify four key challenges (background, biogenic, modelling, and observational) with 18 metrics derived from public datasets, providing a powerful tool for network design and strategic planning.

However, the manuscript's primary weakness is the lack of a clear, reproducible justification for the weighting scheme used to combine the 18 metrics into the four challenge scores. The weights are central to the analysis, yet their derivation is opaquely described as being "assigned based on expert knowledge and consideration of our literature review" without clear descriptions (line 386). This subjectivity undermines the framework's quantitative and objective claims.

Addressing this major point is essential for publication. Additionally, a more in-depth discussion of the study's limitations, particularly regarding city boundary definitions and the selection of metrics, would significantly enhance the scientific rigor, reproducibility, and impact of this important work.

We thank the reviewer for their assessment of our manuscript. Below, we address each of the points raised in blue. Please note that the line numbers in the reviewer's comments refer to the original version of the paper, while our responses pertain to the revised version.

**2. Detailed Comments**

**Weighting of Metrics (Major Point)**

The manuscript's most significant issue is the insufficient justification for the weights assigned to each of the 18 metrics (Tables 1 and 2). The claim that they are based on "expert knowledge" is not sufficient for a scientific paper aiming to establish a quantitative framework. To address this, the authors should:

- **Provide a detailed rationale:** Include an appendix or supplementary material that explains the reasoning for each specific weight. For example, why is "Share of dominant wind" (30%) considered three times more important than "Share of wind >2 m/s" (10%) for the background challenge?
- **Perform a sensitivity analysis:** Ideally, demonstrate or at least give examples of how the final challenge scores and city rankings change under different plausible weighting schemes. This would show the robustness of the conclusions.
- **Discuss alternative methods:** Discuss other, more objective methods for determining weights, even if not implemented (e.g., Principal Component Analysis). If these are too complex, simpler, clearer methods should be considered, or at least suggested. The authors should justify why the current approach was chosen over these alternatives.

**"Provide a detailed rationale"**

The inclusion of the metrics and their individual weights within a challenge are assigned based on our literature review (as presented in Sect. 1) and experience in the field. However, we see no way of quantitatively weighing their relative importance. The connection between monitoring challenges to some metrics appears in several studies. We also have our experience with, for example, the transport model STILT and point sources. Furthermore, we acknowledge the different needs of various stakeholders and the value of local expert knowledge in determining combinations of metrics and weights for their specific challenges.

We see how we can still be clearer and have added section 2.4.1, called "Weights".

Line 392 (2.4.1 Weights): "To reflect that some metrics are expected to contribute more to the challenges than others, they are weighted as specified in Table 1. For the overall challenge, each of the four identified challenges are weighted equally. The individual weights within a challenge are assigned based on our literature review (as presented in Sect. 1) and experience in the field. A sensitivity analysis was performed to assess how the overall challenge score changed under different weighting schemes. Naturally, cities whose metrics almost exclusively indicate that they are either relatively hard or relatively easy to monitor will show more robust challenge scores. Once more data becomes available to link the different metrics to how well an area can be monitored, weights may be assigned in a more quantitative way. However, it should always be possible to adjust choices to accommodate the different needs of stakeholders and to recognize the value of local expert knowledge."

**"Perform a sensitivity analysis"**

The differences in scores for the metrics, regardless of their weights, reflect differences between cities. Cities are bound to rank high on some metrics and low on others, which makes their challenge scores sensitive to the chosen weights. Some cities, whose metrics almost exclusively indicate that they are either relatively hard or relatively easy to monitor, show more robust challenge scores: they will consistently have high or low challenge scores regardless of the weights.

To examine the effect of different weighting schemes across cities, we use a Monte Carlo approach. The relative weights between the different challenges (e.g., biogenic, modelling, background, application-specific observational) are kept equal. The maximum difference between the weights of variables within a challenge is allowed to be up to fourfold. This is consistent with the weighting scheme for the overall challenge used in the study, where the minimum and maximum weights across the 18 metrics were 2.5% and 10%, respectively. The simulation was run 1,000 times.

The figure below shows the results for all cities, positioned along the x-axis according to their mean challenge score across the simulations. Under our chosen weighting scheme for the overall challenge (red star on the y-axis), the French city of Rouen appears to be the most challenging to monitor, while the Dutch city of Leiden lies at the opposite, "relatively easy-to-monitor", end of the spectrum. The same is true in the graph below and means that their mean challenge scores given different weighting schemes yield the same ranking as in the study for these cities.

We have mentioned the sensitivity study in the weights section (see above).

**"Discuss alternative methods"**

We do not see how PCA would be helpful in determining the weighting scheme for the different variables. While PCA might reduce dimensionality by identifying variables that are highly collinear, we intentionally want to retain the compounding effect of two variables pointing in the same direction by giving both of them weights. Furthermore, we believe it is important for readers to be able to trace the challenge scores back to the individual metrics.

Machine learning approaches are not feasible at this stage, given the lack of training data. Such training data would include model—data mismatches, which in turn could be related to different characteristic. However, today we do not see how we could use available data to determine the weight of a single metric, and much less their combination.

We have added a couple of sentences about this to the weighting section (see above), as well as the following to the conclusion (a new final paragraph that also addresses other feedback):

Line 695: "As the field of urban emission monitoring continues to evolve, we anticipate ongoing developments that will both help mitigate current challenges and enhance our suggested methods for analysing them. Regarding the mitigation of challenges, we have highlighted, among other examples, the need for improved urban-specific biogenic models and transport models capable of resolving airflow in urban environments. The research continues to advance, and datasets with higher accuracy than the one used in this study are already available for individual cities. As use of these models is more widely adopted, they can be applied in analyses such as ours. Similarly, with more training data, we may better understand the relationships between different city characteristics and their influence on the ease of monitoring emissions. One possible approach is to use machine learning, correlating model—data differences with these characteristics."

**Methodology & Justification**

• City Definition and Boundaries: The authors correctly note that the OECD functional urban area definition can lead to significant issues, such as excluding major emission sources like the Zurich airport (lines 555-556). The discussion (Section 4) should be expanded to explore the quantitative impact of this. For instance, how much would the "modelling" or "background" challenge scores for Zurich change if the airport were included or excluded? It would also be beneficial to show a brief example of how the proposed "carbon cities" alternative would alter the metrics for one city.

We believe that the administrative boundaries, which form the basis of the OECD boundaries, will continue to be the most relevant framework for cities to use in quantifying their emissions. Additionally, many metrics consider the surrounding areas (20 km buffers) of the cities, such as the emissions with an emphasis of emissions upwind of the city. To highlight this, we have included:

Line 571: "The effect of this is mitigated by the fact that the airport still contributes to the city's challenge scores through its inclusion in the 20-kilometer buffer area (see Sect. 2.1)."

Zurich is highlighted in the text as an extreme case when it comes to borders affecting the results. However, we agree that simulating the inclusion of Zurich's airport serves as a good example of how the score changes depending on the boundaries. The airport falls within the buffer zone around Zurich's border, which means that the only metric affected by the airport being outside the city borders is the "share of point-source emissions". This metric falls under the "modelling challenge", where a higher share of point-source emissions is expected to make monitoring more challenging. The final modelling challenge score for Zurich increased from 54.0 to 61.05, and the corresponding rank changed from 70 to 79.

• **Normalization:** While the 10-90 percentile range for min-max normalization is a reasonable choice to reduce the influence of outliers, the impact of this choice should be briefly stated. For example, how many cities fall outside this range for key metrics, and how does this affect their final scores?

In terms of how many cities fall outside this range, it is always 20% by design. We also want to highlight that they still get a score - just the same score as other cities in the low or high percentile range. We felt this needed clearer explanation in the text and have added:

Line 380: "All cities in the 10th percentile are assigned the value of zero, and those at the 90th percentile are assigned the value of one. The remaining cities are scaled linearly between zero and one."

We have also chosen to remove the equation explaining the normalization (Eq. 1), which we think causes confusion.

In terms of the impact of normalization, the linearly assigned scores reflect how challenging a city is to monitor relative to other cities. In general, the focus should not be on absolute values. If there is a full linear scaling the relative ranking among the cities remain similar, but the challenge scores are lower: the average overall challenge score for all cities would change from 43 (when scaling between the 10th and 90th percentiles) to 37 (when using "full linear scaling"). This makes sense when looking at the outliers in the individual metrics:

In terms of emission intensity in the 20 km buffer zone, Düsseldorf has the highest value, while Ghent, Belgium, has the lowest. Düsseldorf's non-normalized value is 81 times higher than Ghent's. Does this really mean that monitoring in Düsseldorf is 81 times more challenging than in Ghent in this aspect? That would be the influence of the metric in case of full linear scaling. The difference between cities in the 10th and 90th percentiles is also substantial, with the city in the 90th percentile having a value 17 times greater than the city in the 10th percentile.

We have added the following to highlight the connection between the normalization and the final challenge scores:

Line 382: "Without normalisation, an outlier could receive a value of one, while all other cities would get values close to zero. Even after scaling, the metric "Emission intensity buffer" still shows a strong disparity: the city in the 90th percentile has a value 17 times greater than that in the 10th percentile (see Table 2). The effect is evident in Fig. 3b, where the distribution is strongly skewed."

**Specific Challenges & Metrics**

• "Application-Specific Observational Challenge": This category feels heterogeneous, combining challenges for two different measurement systems: satellite (cloud cover) and radiocarbon (nuclear masking/bias). As the authors suggest (lines 614-616), these should be treated as separate challenges. I recommend splitting this into a "Satellite Observation Challenge" and a "Radiocarbon (14C) Challenge" to provide more targeted and actionable information.

The reviewer is referring to the following line: "In the future, considering separate challenge scores for the different observational methods could be beneficial, allowing cities to evaluate their options independently."

**Which we have updated to:**

Line 631: "They can be evaluated independently in Fig. 3, and stakeholders interested in specific cities can consider the two observational methods separately in the mapbooks (Storm et al., 2025b, https://doi.org/10.18160/Z66D-05JT)."

This means that we now describe how the observational methods and their associated challenges can be considered separately. How easy it is to separate the influence also highlights that the final challenge scores are not "black boxes" but can be traced back to the individual metric scores in Fig. 3a for Munich, Zurich, and Paris. A similar figure can be found in the individual mapbooks. Below is the part of Fig. 3 that shows the application-specific observational challenge. It is clear that the contribution to the challenge in Munich, Zurich, and Paris does not come from making radiocarbon measurements but rather from using satellite observations.

It is also worth noting that, as the metrics are equally weighted within the challenge, splitting them would not affect the overall challenge score (assuming that the total weight for the "observational challenge" still adds up to 25%).

• **Biogenic Challenge:** The metrics used are NEE-related and vegetation heterogeneity. However, the influence of urban-specific biogenic factors (e.g., irrigation, urban heat island effects on phenology) is not explicitly captured. The authors should be more explicit about these limitations in the main text when introducing the challenge.

This is also connected to the point further down made specifically about missing metrics. We see how these suggested characteristics would be useful, but are currently not relevant as that sort of data is not available across the 96 cities. We have now stated this general limitation more clearly in Section 2.2 ("Extraction of city metrics")—see below.

In the introductions section (Sect. 1) we bring up how the urban-specific biogenic factors means that the currently used biospheric model in the study is limited (e.g. "However, urban management practices have been shown to violate this assumption. For example, Smith et al. (2019) found urban trees to have growth rates up to four times compared to those observed in a nearby forest....").

We highlight it as an important aspect in mitigating the challenges in the final paragraph of the conclusion:

Line 695: "As the field of urban emission monitoring continues to evolve, we anticipate ongoing developments that will both help mitigate current challenges and enhance our suggested methods for analysing them. Regarding the mitigation of challenges, we have highlighted, among other examples, the need for improved urban-specific biogenic models and transport models capable of resolving airflow in urban environments. The research continues to advance, and datasets with higher accuracy than the one used in this study are already available for individual cities. As use of these models is more widely adopted, they can be applied in analyses such as ours. Similarly, with more training data, we may better understand the relationships between different city characteristics and their influence on the ease of monitoring emissions. One possible approach is to use machine learning, correlating model—data differences with these characteristics."

• Clarification Needed on Wind Speed: The following statement needs clarification: "Higher wind speeds result in larger influence regions ("footprints"), which reduce the impact of strong local sources that could interfere with the goal of obtaining spatially representative observations. This leads to generally better agreement between modeled and observed values"

(Lines 312--313). Is this influence related to the background region? If so, the sentence needs to be clearer. Subtracting a large, uncertain background from an observation can reduce the local signal, making it more uncertain. If the point is about the dilution of the local signal due to high winds, this is a separate issue from the background. These sentences need to be presented more clearly.

We see how this needs clarification and have updated the text:

Line 319: "The challenge of determining the background concentration of CO2 upwind of the city is connected to wind patterns, natural fluxes and anthropogenic emissions. Higher wind speeds result in larger influence regions ("footprints") and reduce the impact of strong local sources within the background region. This leads to more spatially representative background observations and is one reason for excluding low—wind—speed observations from further analyses, such as in the inverse modelling studies over Paris by Bréon et al. (2015) (>2 m s-1) and Lian et al. (2023) (>3 m s-1)."

We have clarified that we are referring to strong local sources in the background region, which create "a large, uncertain background" when wind speeds are low. This serves as a motivation for identifying cities with frequent low wind speeds as having a greater background challenge.

• Missing Metrics: Was the inclusion of other potentially relevant metrics considered? For example, for the "modelling challenge," were metrics for urban canyons or street-level aspect ratios considered? For the "background challenge," was population density in the upwind buffer zone considered as a proxy for diffuse emissions? A brief mention of why certain metrics were considered but ultimately excluded would be helpful.

In terms of "were metrics for urban canyons or street-level aspect ratios considered" we see how this could be useful, but is currently not possible as such a data layer is not available across the 96 cities. We have added the following to section 2.2 ("Extraction of city metrics")

Line 211: "The datasets are available for the entire region, which is a prerequisite for making comparisons across the 96 cities. Alternative datasets and derived metrics—which were excluded from this study—are also available in the notebook tool (Storm et al., 2025a)."

The added final paragraph of the conclusion (see higher up in the document) also highlights how we can and should include more datalayers for metrics as they come available.

In terms of "was population density in the upwind buffer zone considered as a proxy for diffuse emissions", we instead used emissions in the upwind area, which include diffuse emissions distributed based on population density. Including population density as a separate factor would have compounded the effect of upwind diffuse emissions.

**Results & Discussion**

• Results of Munich and Zurich (lines 452--453): While a strong dominant wind direction can help estimate boundary conditions, it may also prevent the sampling of emission sources

across a city unless there are enough sampling sites. The authors do not seem to consider this. It is also very challenging to model emissions for a region with a strong dominant wind direction, as modeled wind biases can lead to significant misestimations, e.g., due to the omission of sources.

The reviewer is referring to the following sentence:

Line 467: "Munich and Zurich both have strong dominant wind directions. This is advantageous for representing the inflow boundary conditions with a limited network of tall tower stations measuring concentrations."

While this is true for the background challenge, we can also see—as pointed out by the reviewer—that wind direction may influence additional challenges related to monitoring emissions. As mentioned in the discussion, "our framework offers a foundation for future discussion and refinement as the research field progresses", and we see this as part of that future discussion. It would be possible to use the tool to include dominant wind direction, with the implication that it increases the difficulty of monitoring, in the modelling challenge. For the "overall challenge", the challenges would offset each other, which highlights the value of considering different types of challenges separately.

In section 2.3.1 (Background challenge) we have added:

Line 331: "This makes the city less challenging to monitor in terms of the background challenge."

• Clarification Needed on Low Wind Speed: The following sentence needs to be explained more clearly: "However, compared to the other cities, the wind speed is quite frequently below 2 m s-1, which could be a challenge for collecting spatially representative up-wind observations." (Lines 453--455).

We have clarified the sentence, and it currently reads as follows:

Line 468: "However, compared to the other cities the wind speed is quite frequently below 2 m s-1. This rather adds to the challenge, as upwind observations are less likely to be spatially representative during periods of low wind-speed."

• **Discussion of Paris (lines 584-599):** The comparison of the paper's findings for Paris with results from Lian et al. (2023) is excellent and provides strong validation for the framework. This section could serve as a model for expanding the discussion of other cities or challenges.

The point refers to the paragraph in the discussion section starting with "Regarding the "biogenic challenge", Lian et al. (2023) highlighted their poorly resolved and non-optimized biogenic fluxes as a key area for improvement in future studies in Paris."

In the paragraph, we highlight how in Paris—despite the biogenic challenge being relatively low—there are still issues and a need for measures to address it. Hence, this provides a lesson that can be applied to other cities as well with similar scores or higher. We agree that this represents strong validation of the framework. For the other challenges, we also reason in a similar way but have adjusted to make the discussion more consistent with the reasoning used in the biogenic-challenge paragraph.

Line 592: "Best practices proposed to mitigate the "background challenge" include using observations to find upwind-downwind gradients for inversions (e.g., Bréon et al., 2015; Staufer et al., 2016), or to constrain the modelled boundary conditions with observations (e.g., Sargent et al., 2018). Our metrics associated with the challenge offer an estimate for how spatially representative the observations may be by considering fluxes nearby the cities. Our consideration of wind speed and direction also ties to how many useful observations would be available for the different practices aimed at limiting the bias from boundary conditions. These factors greatly reduced the number of samples that could be used in the inversion over Paris by Bréon et al. (2015). At the time, the background concentration was sampled from only two towers, and the wind speed threshold, like ours, was 2 m s-1. Compared with the other cities, Paris is among the most favourable in terms of wind speed but exhibits one of the most variable wind directions. Today, more background towers are available, and cities with similar characteristics could also benefit from deploying a larger number of background sites from the outset."

Line 619: "For the "modelling challenge" most of the metrics are related to the complexity of natural and urban topography, which puts high demands on models to accurately resolve the airflow. This is the main driver for Zurich's challenge score (3rd quartile). However, the study by Berchet et al. (2017) conducted in Zurich shows good performance of their model, which they found to fulfil the requirements for air pollution modelling at most of the tested sites. Although the requirement for modelling CO2 is higher, this is promising for cities' abilities to overcome this challenge. Hence, cities with scores similar to that in Zurich could benefit from adopting such a model."

• Conclusion: The conclusion rightly points out that the results are useful for identifying similarities between cities. However, there is little discussion about how to mitigate the identified challenges. Adding a discussion of mitigation strategies would strengthen the paper.

Strategies to mitigate the identified challenges appear throughout the paper, and we highlight a few examples below. However, we agree with the reviewer that it would be useful to connect these points in the conclusion but without excessive repetition, and we have therefore added the following:

Line 695: "As the field of urban emission monitoring continues to evolve, we anticipate ongoing developments that will both help mitigate current challenges and enhance our suggested methods for analysing them. Regarding the mitigation of challenges, we have highlighted, among other examples, the need for improved urban-specific biogenic models and transport models capable of resolving airflow in urban environments. The research continues to advance, and datasets with higher accuracy than the one used in this study are already available for individual cities. As use of these models is more widely adopted, they can be applied in analyses such as ours. Similarly, with more training data, we may better understand the relationships between different city characteristics and their influence on the ease of monitoring emissions. One possible approach is to use machine learning, correlating model—data differences with these characteristics."

**Examples throughout the paper:**

Line 613 (with regards to the biogenic challenge): "Hence, even cities with low scores likely require the use of well-calibrated biospheric models, preferably optimised with complementary direct flux measurements and observations of correlated tracers and/or isotopes. This is likely preferable to the strategy of using observations only in the dormant season (e.g. Lauvaux et al., 2016)."

Line 467 (with regards to the background challenge): "Munich and Zurich both have strong dominant wind directions. This is advantageous for representing the inflow boundary conditions with a limited network of tall tower stations measuring concentrations."

Line 130 (with regards to the modelling challenge): "Another challenge for transport models in the urban environment is to accurately represent airflow, which is complicated by variable topography and tall urban structures. There are models that can do this with some accuracy (e.g. Berchet et al., 2017; Gaudet et al., 2017), but they are computationally expensive to run. For example, Berchet et al. (2017) use a catalogue-based approach where a set of pre-computed steady-state flow and dispersion patterns is matched hourly to actual meteorological observations."

Line 638 (with regards to the application-specific observational challenge): "For the use of 14CO2 observations, the observational challenge is linked to how much contribution is expected from emissions from nuclear facilities. As in previous studies (e.g., Maier et al., 2023), we used a flat annual emission rate to simulate this, but improving the resolution of emission data is a priority at the ICOS Radiocarbon Laboratory. For example, knowing the timing of emissions from La Hague, France, would significantly enhance the feasibility of using radiocarbon in many cities beyond those closest to it."

**Presentation & Clarity**

• Figure 3 (Challenge Scores): This is an effective figure. A minor suggestion for panel (a) is to add the numerical score for each of the three pilot cities next to their bars to make comparison with Table 3 easier.

We thank the reviewer for the suggestion, but we prefer not to include numbers on the individual bars. We believe the figure is already quite busy. Bars belonging to the same challenge are close together and can be compared easily. For comparisons across different challenges, Table 3 can be consulted. Similarly, the mapbooks for the individual cities provide this information in a table next to the figure.

• Table 1: This table could be improved with some explanatory comments. For example, an asterisk could note that the weights within each category sum to 1. The meaning of some metrics is not immediately clear from the table, such as "Share of wind from dominant wind direction (limited to >2 m s-1)". It is also not clear if a higher or lower share is better and why (after reading the paper more, it is better, but providing more information at the beginning would help the reader).

The suggestions have been incorporated in Table 1 (and all other times the metrics are mentioned) and its caption.

Previous:

"Share of wind from dominant wind direction (limited to >2 m s-1)"

Updated:

"Fraction of time wind from the dominant wind direction (limited to wind speed >2 m s-1)"

Added to the caption: "The weights within each category sum to 100%."

To already make it clearer in Table 1 that a higher value means a greater monitoring challenge, we have updated the following to the caption (asterisk (\*) included for metrics that need to be inverted to ensure a higher value indicate a greater monitoring challenge):

**Previous:**

\*Values are inverted to make a higher value mean a greater monitoring challenge (see Sect 2.4)."

**Updated:**

\*For these contributions, a lower value means a greater monitoring challenge (see Sect 2.4)."

We have also added the following to the section where Table 1 is displayed:

Line 215: "Finally, sections 2.4.1 through 2.4.4 outlines how the metrics are integrated and analysed. This includes how the weights (column "Challenge (weight)" in Table 1) are applied to the individual metrics, and how the metrics are adjusted so that higher values consistently correspond to a greater monitoring challenge before they are combined."

• Metric Explanations: Does Section 2.2 explain all the metrics in Table 1? The section seems to mix data source and metric descriptions, and some metrics appear to be missing a clear explanation.

Our intention was indeed to introduce the metrics, but with a focus on those that cannot be understood from their name alone (e.g., "Share point source emission"). However, we recognize the importance of ensuring consistency and not omitting any metrics.

Line 242 (section 2.2.2): "All metrics related strictly to anthropogenic CO2 emissions use the annual total for the year 2018. "Emission intensity buffer" uses emissions per km2 in the 20-kilometer buffer area around the cities, with an additional metric, "Emission intensity buffer dominant wind direction", which highlights the upwind buffer area. "Share point source emissions" is the percentage of a city's total emissions that can be attributed to point sources."

Line 269 (section 2.2.3): "The ESA Worldcover dataset is also used to include cropland information (class 40) in the metrics "share cropland buffer" and "share cropland in dominant wind direction". In both cases, 20-kilometer buffers around the cities are applied (see Sect. 2.1). For "share cropland buffer", the full buffer area is used. For "share cropland in dominant wind direction", only the buffer in the dominant wind direction is considered to emphasise the upwind area."

**• Equations:**

Equation 1: The term x\_{i,j} is not explained. The index j represents
"characteristics," but the paper states, "These metrics represent specific characteristics of the city" (Line 158), implying that characteristics are the metrics. This should be clarified, although it appears to be.

The reviewer is referring to equation 2, not equation 1. See previous discussion on how this equation is removed from Sect 2.4 and replaced by text to explain how the values are normalized.

We see how "characteristics" and "metrics" have not been used consistently throughout the paper and have made updates where needed. "These metrics represent specific characteristics of the city" is correct.

• Equation 3: This equation needs more explanation. The term x\_i (or y\_i) appears to be a vector. How is it formed? For example, for the background challenge, does the vector x have 5 elements? This is not clear from the text, although it seems so.

We have made clarifications to the equation (now Equation 2):

"The scaled and weighted characteristics are used to create similarity matrices based on Euclidean distances. The Euclidean distance, D, between two cities x and y is calculated as follows:

$$D(x,y) = \sqrt{\sum_{i=1}^{n} (x_i - y_i)^2}$$
 (2)

Where  $x_i$  and  $y_i$  represent the *i*th scaled and weighted metrics for cities x and y respectively. The resulting distances are summarized across the individual challenges as well as for the overall challenge. All cities are compared to all others forming 96x96 matrices."

• **Typos and Grammar:** There are several typos (e.g., "with an emphasize on the surrounding" in Line 228) and long sentences that could be revised to improve readability.

We have read through and corrected typos and grammatical errors, including the one pointed out by the reviewer.